# FIT-GNN: Faster Inference Time for GNNs that 'FIT' in Memory Using Coarsening

**Shubhajit Roy**[*]                                    *royshubhajit@iitgn.ac.in*
*Indian Institute of Technology Gandhinagar*

**Hrriday Ruparel**[*]                                  *hrriday.ruparel@iitgn.ac.in*
*Indian Institute of Technology Gandhinagar*

**Kishan Ved**                                          *kishan.ved@iitgn.ac.in*
*Indian Institute of Technology Gandhinagar*

**Anirban Dasgupta**                                    *anirbandg@iitgn.ac.in*
*Indian Institute of Technology Gandhinagar*

**Reviewed on OpenReview:** *https://openreview.net/forum?id=g7r7y2I7Sz*

## Abstract

Scalability of Graph Neural Networks (GNNs) remains a significant challenge. To tackle this, methods like coarsening, condensation, and computation trees are used to train on a smaller graph, resulting in faster computation. Nonetheless, prior research has not adequately addressed the computational costs during the inference phase. This paper presents a novel approach to improve the scalability of GNNs by reducing computational burden during the inference phase using graph coarsening. We demonstrate two different methods – Extra Nodes and Cluster Nodes. Our study extends the application of graph coarsening for graph-level tasks, including graph classification and graph regression. We conduct extensive experiments on multiple benchmark datasets to evaluate the performance of our approach. Our results show that the proposed method achieves orders of magnitude improvements in single-node inference time compared to traditional approaches. Furthermore, it significantly reduces memory consumption for node and graph classification and regression tasks, enabling efficient training and inference on low-resource devices where conventional methods are impractical. Notably, these computational advantages are achieved while maintaining competitive performance relative to baseline models.

## 1 Introduction

Graph Neural Networks (GNNs) have demonstrated remarkable versatility and modeling capabilities. However, several challenges still hinder their widespread applicability, with scalability being the most significant concern. This scalability issue affects both the training and inference phases. Previous research has sought to mitigate the high training computation cost by using three broad categories of methods: 1. Training every iteration on a sampled subgraph 2. Training on a coarsened graph generated from a coarsening algorithm 3. Training on a synthetic graph that mimics the original graph. Unfortunately, these smaller graphs can only be utilized for training purposes and have not alleviated the computational burden during inference since the method mentioned above requires the whole graph during the inference phase. Consequently, for larger graphs, not only do inference times increase, but memory consumption may also exceed limitations.

To mitigate the high computational cost associated with the inference phase, we decompose the original graph $G$ into a collection of subgraphs $\mathcal{G}_s = \{G_1, G_2, \ldots, G_k\}$. This transformation yields substantial

---

[*]Equal contribution

improvements in inference efficiency, achieving up to a **100×** speedup for single-node prediction. Moreover, it significantly reduces memory consumption by as much as **100×** —thereby enabling scalable inference on large-scale graphs comprising millions of edges. We provide a theoretical characterization of the upper bound on the expected size of subgraphs under which our method's time and space complexity remain superior to that of the baselines. Notably, we observe that conventional methods often fail to perform inference on large graphs due to memory constraints, whereas our approach remains effective and scalable in such scenarios.

While partitioning the graphs into subgraphs seems to be a naive transformation, it comes with the cost of information loss corresponding to the edges removed. We overcome this loss by appending additional nodes in the subgraphs using two different methods – **Extra Nodes** and **Cluster Nodes** that respectively append neighboring nodes or representational nodes corresponding to neighboring clusters. These additional nodes resulted in equivalent or better performance in terms of accuracy and error compared to baselines. Our contributions are summarized as follows:

1. We reduce inference time and memory by applying graph coarsening to partition graphs into subgraphs, mitigating information loss through **Extra Node** and **Cluster Node** strategies.

2. We provide a theoretical analysis of our approach, establishing a lower inference computational complexity compared to baselines.

3. Our method scales well to extremely large graphs that baselines fail to process, while consistently delivering strong performance on 13 real-world datasets.

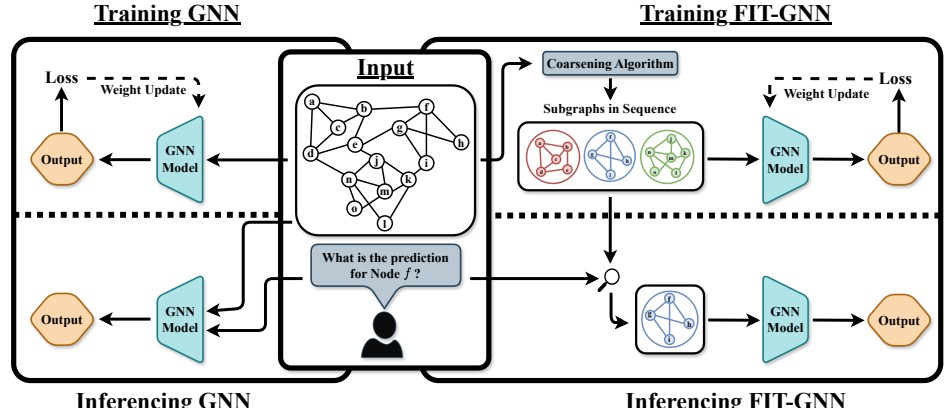

Figure 1: The Figure shows the overall pipeline of our proposed method and its comparison with traditional training and inference of GNNs. This pipeline is made for node-level tasks.

## 2  Related Work

The aim of reducing computational costs has been approached through various methods. Research by Chiang et al. (2019) and Zeng et al. (2020) uses a subgraph sampling technique, where a small subgraph is sampled at each training iteration for GNN training. Other methods, including layer-wise sampling and mini-batch training, have been explored by Chen et al. (2018a;b); Cong et al. (2020); Zou et al. (2019). Additionally, Fahrbach et al. (2020) introduced a coarsening algorithm based on Schur complements to create node embeddings in large graphs, while Huang et al. (2021) [**SGGC**] applied GNNs to coarsened graphs, reducing nodes and edges to lower training time. Kumar et al. (2023); Joly & Keriven (2024); Kataria et al. (2024); Dickens et al. (2024) also proposed different coarsening algorithms for scalable training approaches. Jin et al. (2021) [**GCOND**], Wang et al. (2024) [**GC-SNTK**] presented algorithms to generate a synthetic graph that mimics the original graph. However, their approach generates model-specific synthetic graphs that require training on the whole dataset, which defeats the purpose of reducing computational burden. Gupta et al. (2025) [**BONSAI**] addressed this issue by looking into computational trees and training on just the relevant computational trees. However, these approaches do not decrease inference time since the inference phase takes the whole graph as input. Xue et al. (2023) shifted the training process to subgraph-level, using coarsening algorithms to split the graph into $k$-subgraphs and recommending a multi-device

training approach. In the graph-level task, Jin et al. (2022) [**DOSCOND**] and Xu et al. (2023) [**KIDD**] tackled the graph classification problem following up on the approach of GCOND and GC-SNTK. However, the graph regression problem was not explored in this direction.

## 3 Preliminaries

**Graph Notations:** Given an undirected graph $G = (V, E, X, W)$, $V$ is the vertex set, $E$ is the edge set, $X \in \mathbb{R}^{n \times d}$ is the feature matrix, and $W$ is the edge weight matrix. Let $|V| = n$ be the number of nodes and $|E| = m$ be the number of edges. Let $A \in \mathbb{R}^{n \times n}$ be the adjacency matrix of $G$ where $A_{ij}$ is the edge weight between nodes $v_i$ and $v_j$, and $d_i$ is the degree of the node $v_i$. $D \in \mathbb{R}^{n \times n}$ is a diagonal degree matrix with $i$th diagonal element as $d_i$. We denote $\mathcal{N}_j(v_i)$ as the $j$-hop neighbourhood of node $v_i \in V$. **Graph Neural Network:** Graph Neural Network is a neural network designed to work for graph-structured data. The representation of each node in the graph is updated recursively by aggregation and transforming the node representations of its neighbors. Kipf & Welling (2017) introduced Graph Convolutional Network (GCN) as follows:

$$X^{(l+1)} = \sigma(\tilde{D}^{-\frac{1}{2}} \tilde{A} \tilde{D}^{-\frac{1}{2}} X^{(l)} \mathcal{W}^{(l)}) \tag{1}$$

where $X^{(l)}$ is the node representation after $l$ layers, $\tilde{A} = A + I$, $\tilde{D} = D + I$, $I$ is the identity matrix of same dimension as $A$. $\mathcal{W}^{(l)}$ is the learnable parameter and $\sigma$ is a non-linear activation function.

**Graph Coarsening and Graph Partitioning:** Loukas (2019) discussed multiple coarsening algorithms which create a coarsened graph $G' = (V', E', X', W')$ from a given graph $G = (V, E, X, W)$. We refer the vertex set $V' = \{v'_1, v'_2, \ldots, v'_k\}$ of $G'$ as *coarsened nodes*. Given a coarsening ratio $r \in [0, 1]$, we have $k = \lfloor n \times r \rfloor$, $k \in \mathbb{Z}$. We can interpret it as creating $k$ disjoint partitions, $C_1, C_2, \ldots, C_k$, from a graph of $n$ nodes. Mathematically, we create a partition matrix $P \in \{0, 1\}^{n \times k}$, where $P_{ij} = 1$ if and only if node $v_i \in C_j$. $X' = P^\top X$ is the coarsened node representation. $A' = P^\top A P$ is the adjacency matrix of $G'$. Similarly, the corresponding degree matrix $D' = P^\top D P$. SGGC (Huang et al., 2021) used normalized partition matrix $\mathcal{P} = PC^{-\frac{1}{2}}$, where $C$ is defined as a diagonal matrix with diagonal entries $C_{jj} = |C_j|, j = 1, 2, \ldots, k$.

Along with $G'$, we create a set of disjoint subgraphs $\mathcal{G}_s = \{G_1, G_2, \ldots, G_k\}$ corresponding to the partitions $C_1, C_2, \ldots, C_k$. The number of nodes in each partition $C_i$ is denoted by $n_i$. Each subgraph $G_i$ is the induced subgraph of $G$ formed by the nodes in $C_i$. $A_i$ and $D_i$ are the adjacency and degree matrix of $G_i$ respectively.

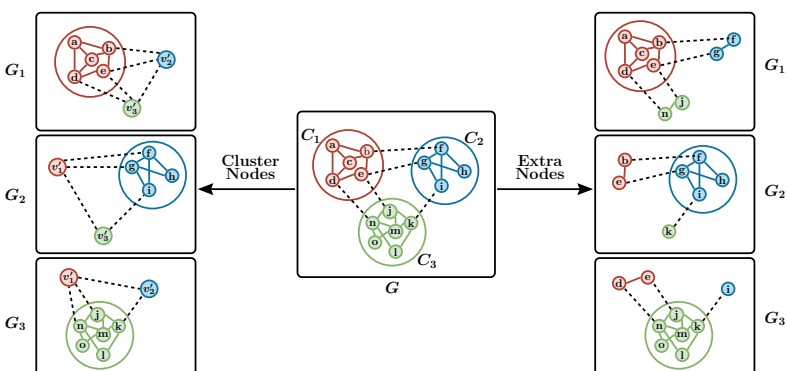

Figure 2: Figure showing the comparison between the Extra Node Method and the Cluster Node Method of appending additional nodes in $G_1, G_2, G_3$.

## 4 Methodology

In SGGC, a graph $G$ is reduced to $G'$ using a partition matrix $P$ generated from the coarsening algorithm. The labels for the coarsened graph are $Y' = \arg\max(P^\top Y)$, where $Y$ is the label matrix of $G$ storing the one-hot encoding of the label of each node for the node classification task. Training is carried out on

the coarsened graph $G'$ using $Y'$ as the label information. While this approach is novel, essential label information is lost. For instance, suppose a node $v' \in G'$ is a combination of $v_i, v_j, v_k \in G$ where $v_i, v_j, v_k$ are test nodes. Let us assume that the classes of $v_i, v_j, v_k$ are $0, 0, 2$ respectively. $\arg\max$ will take the majority label and assign class 0 to $v'$. Hence, the model will be trained to predict only the class 0, leading to the discarding of the quantification of the model's performance on predicting less represented nodes.

Similarly, to explore another node-level task, such as node regression, aggregation functions such as the mean of regression targets wouldn't be appropriate because they lose the variance of the targets. To mitigate label information loss, we use original label information without any aggregation function like $\arg\max$ or mean, thus motivating us to follow a subgraph-level training process.

However, partitioning the graph into subgraphs leads to a loss of neighborhood information at the periphery of subgraphs. To tackle this, we append nodes in two ways after partitioning:

- **Extra Nodes:** Adding the 1-hop neighbouring nodes, namely Extra Nodes, in each subgraph $G_i$ denoted by $\mathcal{E}_{G_i}$. Xue et al. (2023) introduced this approach to reduce information loss after partitioning. We add a unit weight edge if two nodes in $\mathcal{E}_{G_i}$ are connected in $G$. Node embedding of $v \in \mathcal{E}_{G_i}$ is $x_v \in X$.

$$\mathcal{E}_{G_i} = \bigcup_{v \in G_i} \{u \in V : u \in \mathcal{N}_1(v) \wedge u \notin G_i\} \tag{2}$$

- **Cluster Nodes:** Although adding 1-hop neighboring nodes in the subgraph helps recover lost information after partition, for multi-layered GNN models, information loss will persist. Instead of adding the neighboring nodes, we will add Cluster Nodes, which represent the neighboring clusters, denoted by $\mathcal{C}_{G_i}$. Liu et al. (2024) introduced this method to add nodes in the graph instead of adding to separate subgraphs. They also added cross-cluster edges among these Cluster Nodes and proposed a subgraph sampling-based method on the modified graph. In our work, we add cross-cluster edges. For each node $t \in \mathcal{C}_{G_i}$, the node embedding $x_t = X'_t$.

$$\mathcal{C}_{G_i} = \bigcup_{v \in \mathcal{E}_{G_i}} \{t : P_{v,t} \neq 0\} \tag{3}$$

Figure 2 illustrates appending these additional nodes using these approaches. The dashed line shows which nodes are added as part of these additional nodes. For subgraph-level training, specifically for node-level tasks, the newly appended nodes do not contribute to the weight update as the loss is not backpropagated based on the predictions of these nodes. This is done since the appended nodes might contain information from the test nodes. Therefore, a boolean array $mask_i$ is created for each subgraph $G_i$ such that $True$ is set for nodes that actually belong to the subgraph $G_i$ (not as an Extra Node or Cluster Node) and are training nodes; otherwise, $False$.

**Lemma 4.1.** *Models with 1 layer of GNN cannot distinguish between $G$ and $\mathcal{G}_s$ when **Extra Nodes** method is used. (Proof in Appendix A.1)*

Using the **Cluster Nodes** approach to append additional nodes to subgraphs is better for two reasons:

- From Equation 2 and 3, we can say that $\sum_i |\mathcal{E}_{G_i}| \geq \sum_i |\mathcal{C}_{G_i}|$. This is because, using Extra Nodes, we add all the neighboring nodes. However, using Cluster Nodes, we only add one representative node for each neighboring cluster. Hence, the time taken to train and infer these modified subgraphs with Cluster Nodes is at most that with Extra Nodes.

- Lemma 4.1 shows that the Extra Nodes approach reduces the information loss due to partitioning for 1 Layer of GNN. However, with multiple layers of GNN, longer dependencies are not captured. In contrast, the Cluster Nodes approach overcomes this by computing a single Cluster Node's representation as a combination of features of all the nodes present in the corresponding cluster. This results in sharing further node information in 1-hop. Additionally, the transfer of information from one cluster to another is captured.

---

**Algorithm 1** Training GNN (Train on $\mathcal{G}_s$)

---

**Require:** $G = (V, E, X)$; Labels $Y$; Node-Model $M$; Loss $\ell$; Number of Layers $L$
  1: Apply a coarsening algorithm on $G$, and output a normalized partition matrix $P$.
  2: Construct $\mathcal{G}_s = \{G_1(V_1, E_1, X_1), G_2(V_2, E_2, X_2), \ldots, G_k(V_k, E_k, X_k)\}$ using $P$;
  3: **if** Append additional nodes **then**
  4:     Append $\mathcal{E}_{G_i}$ or $\mathcal{C}_{G_i}$ to each $G_i$                                             ▷ Based on method of appending nodes
  5: **end if**
  6: Construct mask for each subgraph $Mask = \{mask_1, mask_2, \ldots, mask_k\}$;
  7: Construct label matrix $\{Y_1, Y_2, \ldots, Y_k\}$ for $\mathcal{G}_s$ ;
  8: $O \leftarrow []$; $Y_{\text{Collected}} \leftarrow []$;
  9: **for** each subgraph $G_i$ in $\mathcal{G}_s$ **do**
 10:     $O_i = M(L, A_i, D_i, X_i)$;                                                          ▷ Algorithm 4
 11:     $O.append(O_i[mask_i])$;
 12:     $Y_{\text{Collected}}.append(Y_i[mask_i])$;
 13: **end for**
 14: $Loss = \ell(O, Y_{\text{Collected}})$;
 15: Train $M$ to minimize $Loss$;

---

## 4.1 Node-level Task

We use Algorithm 4 with $L$ number of layers to create a standard GNN model $M$ for node-level tasks. While we use Algorithm 3 to train a model on the coarsened graph $G'$, we propose Algorithm 1 to train a model on the set of subgraphs $\mathcal{G}_s$. Algorithm 3 utilizes the partition matrix $P$ generated by the coarsening algorithm to construct and train on the coarsened graph $G'$ and the coarsened label information $Y'$. On the other hand, our proposed Algorithm 1 uses the same partition matrix $P$ to construct and train on subgraphs $\mathcal{G}_s$ using original label information $Y$. For the node classification task, $\arg\max$ is used for creating $Y'$ for $G'$ and CrossEntropy as a loss function. For the node regression task, we do not create $G'$. Mean Absolute Error (MAE) is used as the loss function.

## 4.2 Graph-level Task

For a given graph $G \in \mathcal{D}$, we create $G'$ and $\mathcal{G}_s = \{G_1, G_2, \ldots, G_k\}$. Since $G'$ is a single graph and $\mathcal{G}_s$ is a set of subgraphs, we use Algorithm 5 to create a model that trains on $G'$ and Algorithm 2 to create a model that trains on $\mathcal{G}_s$. We use the CrossEntropy loss function for the graph classification task and the Mean Absolute Error (MAE) for the graph regression task.

---

**Algorithm 2** Graph-Model-$\mathcal{G}_s(\mathcal{G}_s, L)$

---

**Require:** $\mathcal{G}_s = \{G_1, G_2, \ldots, G_k\}$; Number of Layers $L$;
  1: $\tilde{X} \leftarrow []$
  2: **for** each subgraph $G_i \in \mathcal{G}_s$ **do**
  3:     **for** $j = 1$ to $L$ **do**
  4:         $X_i^{(j)} = \sigma(\tilde{D}_i^{-\frac{1}{2}} \tilde{A}_i \tilde{D}_i^{-\frac{1}{2}} X_i^{(j-1)} \mathcal{W}^{(j-1)})$                                    ▷ Equation 1
  5:     **end for**
  6:     $\tilde{X}.stack(X_i^{(L)})$                                                  ▷ Stack along row axis
  7: **end for**
  8: $\bar{X} = \text{MaxPooling}(\tilde{X})$
  9: $Z = \bar{X} \mathcal{W}^{(L)}$
 10: **return** $Z$

---

## 4.3 Time and Space Complexity

In Equation 1, the matrix dimensions are as follows: $\tilde{A}$ has dimensions $(n \times n)$, $X^{(l)}$ is of dimensions $(n \times d)$, and $\mathcal{W}^{(l)}$ has dimensions $(d \times d)$. Time complexity for one layer of GNN computation is $\mathcal{O}(n^2 d + n d^2)$. If we take $L$ layers, then the total time is $\mathcal{O}(Ln^2 d + Ln d^2)$. The space complexity is $\mathcal{O}(n^2 + Lnd + Ld^2)$. When we compute on a sparse graph, the time complexity is $\mathcal{O}(m + Lnd^2)$ and space complexity is $\mathcal{O}(m + Lnd + Ld^2)$. The complexity is based on GCN. For other architectures like GAT (Veličković et al., 2018), the complexity changes.

Table 1: Training and inference time and space complexity of FIT-GNN compared to classical and SGGC approaches.

| (a) Time Complexity | | | | (b) Space Complexity | | |
|---|---|---|---|---|---|---|
| | **Train** | **Inference** | | | **Train** | **Inference** |
| **Classical** | $nd^2 + n^2d$ | $nd^2 + n^2d$ | **Classical** | | $n^2 + nd + d^2$ | $n^2 + nd + d^2$ |
| **SGGC** | $kd^2 + k^2d$ | $nd^2 + n^2d$ | **SGGC** | | $k^2 + kd + d^2$ | $n^2 + nd + d^2$ |
| **FIT-GNN** | $kd^2 + k^2d$ $+\sum_{i=1}^{k}[\bar{n}_i^2 d + \bar{n}_i d^2]$ | $\sum_{i=1}^{k}[\bar{n}_i^2 d + \bar{n}_i d^2]$ | **FIT-GNN** | | $k^2 + kd + d^2$ $+ \max_{i=1}[\bar{n}_i^2 + \bar{n}_i d]$ | $d^2 + \max_{i=1}[\bar{n}_i^2 + \bar{n}_i d]$ |

SGGC improved the time and space complexity (Table 1) for training the network by reducing the number of nodes from $n$ to $k$. However, the inference time and space complexity remain the same.

Let us compare three different models. One is the classical model, where no coarsening or partitioning is done; the second is SGGC, where training is performed on a smaller graph $G'$ and inference on $G$; the third is our approach, FIT-GNN, where both training and inference are performed on $\mathcal{G}_s$.

The inference time complexity of our model is $\mathcal{O}\left(\sum_{i=1}^{k}[\bar{n}_i^2 d + \bar{n}_i d^2]\right)$, where $\bar{n}_i = n_i + \phi_i$ and $\phi_i$ is the number of additional nodes appended in each subgraph. Table 1 compares different approaches.

**Lemma 4.2.** *The inference time complexity* $\mathcal{O}\left(\sum_{i=1}^{k}[(n_i + \phi_i)^2 d + (n_i + \phi_i)d^2]\right)$ *is at most* $\mathcal{O}\left(n^2 d + nd^2\right)$ *if* $\mathbb{E}[n_i + \phi_i] \leq \sqrt{\frac{d^2}{4} + \frac{d}{r} + \frac{n}{r} - \text{Var}(n_i + \phi_i)} - \frac{d}{2}$, *where* $\mathbb{E}[n_i + \phi_i]$ *is the expected number of nodes in each modified subgraph and* $\text{Var}(n_i + \phi_i)$ *is the variance of number nodes in each modified subgraph.*

*Proof.* We can expand our inference time complexity as follows:

$$\sum_{i=1}^{k}[(n_i + \phi_i)^2 d + (n_i + \phi_i)d^2] = nd^2 + d\left[\sum_{i=1}^{k} n_i^2 + \sum_{i=1}^{k} \phi_i^2 + 2\sum_{i=1}^{k} n_i\phi_i + d\sum_{i=1}^{k} \phi_i\right]$$

Now, define expected number of nodes in each subgraph $\mathbb{E}[n_i] := \frac{1}{k}\sum_{i=1}^{k} n_i = \frac{n}{k} = \frac{1}{r}$ (since $k = nr$) and expected number of additional nodes appended in each subgraph $\mathbb{E}[\phi_i] := \frac{1}{k}\sum_{i=1}^{k} \phi_i$. We also define the variances and covariance,

$$\text{Var}(n_i) := \frac{1}{k}\sum_{i=1}^{k} n_i^2 - \mathbb{E}[n_i]^2, \quad \text{Var}(\phi_i) := \frac{1}{k}\sum_{i=1}^{k} \phi_i^2 - \mathbb{E}[\phi_i]^2, \quad \text{Cov}(n_i, \phi_i) = \text{Cov} := \frac{1}{k}\sum_{i=1}^{k} \phi_i n_i - \mathbb{E}[\phi_i]\mathbb{E}[n_i]$$

We can also write,

$$\text{Var}(n_i + \phi_i) = \frac{1}{k}\sum_{i=1}^{k}(n_i + \phi_i)^2 - (\mathbb{E}[n_i] + \mathbb{E}[\phi_i])^2 = \text{Var}(n_i) + \text{Var}(\phi_i) + 2\text{Cov}(n_i, \phi_i)$$

Using the definitions above, we can say:

$$\sum_{i=1}^{k}[(n_i + \phi_i)^2 d + (n_i + \phi_i)d^2] = nd^2 + d\left[\sum_{i=1}^{k} n_i^2 + \sum_{i=1}^{k} \phi_i^2 + 2\sum_{i=1}^{k} n_i\phi_i + d\sum_{i=1}^{k} \phi_i\right]$$

$$= nd^2 + d\left[k(\text{Var}(n_i) + \mathbb{E}[n_i]^2) + k(\text{Var}(\phi_i) + \mathbb{E}[\phi_i]^2) + 2k(\text{Cov}(n_i, \phi_i) + \mathbb{E}[n_i]\mathbb{E}[\phi_i]) + dk\mathbb{E}[\phi_i]\right]$$

$$= nd^2 + nd\left[r\text{Var}(n_i) + \frac{1}{r} + r\text{Var}(\phi_i) + r\mathbb{E}[\phi_i]^2 + 2r\text{Cov}(n_i, \phi_i) + 2\mathbb{E}[\phi_i] + rd\mathbb{E}[\phi_i]\right]$$

$$= nd^2 + nd\left[r\mathbb{E}[\phi_i]^2 + (2 + rd)\mathbb{E}[\phi_i] + r\text{Var}(n_i + \phi_i) + \frac{1}{r}\right]$$

Given, $\mathbb{E}[n_i + \phi_i] \leq \sqrt{\frac{d^2}{4} + \frac{d}{r} + \frac{n}{r} - \text{Var}(n_i + \phi_i)} - \frac{d}{2} \Rightarrow \mathbb{E}[\phi_i] \leq \sqrt{\frac{d^2}{4} + \frac{d}{r} + \frac{n}{r} - \text{Var}(n_i + \phi_i)} - (\frac{d}{2} + \frac{1}{r})$.
Using this, we can say the following:

$$\sum_{i=1}^{k}[(n_i+\phi_i)^2 d + (n_i + \phi_i)d^2] = nd^2 + nd\left[r\mathbb{E}[\phi_i]^2 + (2 + rd)\mathbb{E}[\phi_i] + r\text{Var}(n_i + \phi_i) + \frac{1}{r}\right]$$

$$\leq nd^2 + nd\left[r\left(\frac{d^2}{4} + \frac{d}{r} + \frac{n}{r} - \text{Var}(n_i + \phi_i) + \frac{d^2}{4} + \frac{1}{r^2} + \frac{d}{r} - 2\left(\frac{d}{2} + \frac{1}{r}\right)\sqrt{\Delta}\right)\right.$$

$$\left. +2\sqrt{\Delta} - 2\left(\frac{d}{2} + \frac{1}{r}\right) + rd\sqrt{\Delta} - rd\left(\frac{d}{2} + \frac{1}{r}\right) + r\text{Var}(n_i + \phi_i) + \frac{1}{r}\right]$$

$$\left(\text{where } \frac{d^2}{4} + \frac{d}{r} + \frac{n}{r} - \text{Var}(n_i + \phi_i) = \Delta\right)$$

$$= nd^2 + n^2 d$$

Hence, $\sum_{i=1}^{k}[(n_i + \phi_i)^2 d + (n_i + \phi_i)d^2] \leq n^2 d + nd^2$. $\qquad\square$

**Corollary 4.3.** $\mathbb{E}[\phi_i]$ *has a positive upper bound when* $\text{Var}(n_i + \phi_i) \leq \frac{n}{r} - \frac{1}{r^2}$.

*Proof.* From Lemma 4.2, we know the upper bound of $\mathbb{E}[\phi_i]$ is $\sqrt{\frac{d^2}{4} + \frac{d}{r} + \frac{n}{r} - \text{Var}(n_i + \phi_i)} - \left(\frac{d}{2} + \frac{1}{r}\right)$.

If $\sqrt{\frac{d^2}{4} + \frac{d}{r} + \frac{n}{r} - \text{Var}(n_i + \phi_i)} - \left(\frac{d}{2} + \frac{1}{r}\right) \geq 0$, then we can say the following:

$$\sqrt{\frac{d^2}{4} + \frac{d}{r} + \frac{n}{r} - \text{Var}(n_i + \phi_i)} - \left(\frac{d}{2} + \frac{1}{r}\right) \geq 0 \Rightarrow \frac{d^2}{4} + \frac{d}{r} + \frac{n}{r} - \text{Var}(n_i + \phi_i) \geq \left(\frac{d}{2} + \frac{1}{r}\right)^2$$

$$\Rightarrow \text{Var}(n_i + \phi_i) \leq \frac{n}{r} - \frac{1}{r^2}$$

$\qquad\square$

For a given graph, when the above conditions are satisfied, our approach has better inference time and space complexity than other approaches. From Corollary 4.3, we can understand that it is ideal to have similarly sized subgraphs. For more details on time and space complexity, refer to Section C in the Appendix.

## 5 Experiments

Let training on the coarsened graph $G'$ be referred to as **Gc-train**, subgraph-level training as **Gs-train**, and subgraph-level inference process as **Gs-infer**. Once we have constructed $G'$ and $\mathcal{G}_s$, we train FIT-GNN in 4 different setups:

- **Gc-train-to-Gs-train**: Train the GNN model on $G'$ as per Algorithm 3, then use the learned weight as an initialization for subgraph-level training and final inference on $\mathcal{G}_s$. We examine whether pretraining on $G'$ (which requires less time) followed by fine-tuning on $\mathcal{G}_s$ for fewer epochs can improve performance.

- **Gc-train-to-Gs-infer**: Train the GNN model on $G'$ as per Algorithm 3, then infer on $\mathcal{G}_s$ using the learned weights. We explore whether training solely on the $G'$ is sufficient for inference on $\mathcal{G}_s$. This setting presents a computationally efficient option, allowing users with limited resources to obtain results close to the best reported.

- **Gs-train-to-Gs-infer**: Perform subgraph-level training and inference on $\mathcal{G}_s$.

- **Gc-train-to-Gc-infer**: Unlike node-level tasks, for graph-level tasks, inference can be done directly on $G'$ because the label corresponds to the entire graph and not to individual nodes. Therefore, even with a reduced graph representation, meaningful inference is possible. Therefore, in this setup for graph-level tasks only, we train and infer on $G'$.

All four setups can be applied to graph-level tasks. For the node classification task, we apply all setups except **Gc-train-to-Gc-infer**. For the node regression task, we only perform **Gs-train-to-Gs-infer** because a coarsened graph is not created for node regression datasets.

Dataset descriptions and details related to hyperparameters, device configuration, and key packages used for the experiments are in Section D and E. Source code: $\mathbf{\Omega}$ https://github.com/Roy-Shubhajit/FIT-GNN.

Table 2: Datasets and corresponding splits used in our experiments. 'c' denotes the number of classes in node classification datasets. Public split "fixed" by Yang et al. (2016) is used for *Cora*, *Citeseer*, and *PubMed*. For the "random" split, we use 20 nodes per class for training, 30 per class for validation, and the rest for testing. For *QM9*, we follow Gilmer et al. (2017) and predict one property from each broad category: dipole moment ($\mu$), HOMO-LUMO gap ($\epsilon_{\text{HOMO}} - \epsilon_{\text{LUMO}}$), zero-point vibration energy (ZPVE), and atomization energy ($U^{\text{ATOM}}$) at 298.15K.

| Task | Dataset (Reference) | Train | Val | Test |
|------|--------------------|-------|-----|------|
| **Node Regression** | *Chameleon* (Rozemberczki et al., 2021) | 30% | 20% | 50% |
| | *Crocodile* (Rozemberczki et al., 2021) | 30% | 20% | 50% |
| | *Squirrel* (Rozemberczki et al., 2021) | 30% | 20% | 50% |
| **Node Classification** | *Cora* (McCallum et al., 2000) | $20 \times c$ | $30 \times c$ | Rest |
| | *Citeseer* (Giles et al., 1998) | $20 \times c$ | $30 \times c$ | Rest |
| | *Pubmed* (Sen et al., 2008) | $20 \times c$ | $30 \times c$ | Rest |
| | *Coauthor Physics* (Shchur et al., 2018) | $20 \times c$ | $30 \times c$ | Rest |
| | *DBLP* (Tang et al., 2008) | $20 \times c$ | $30 \times c$ | Rest |
| | *OGBN-Products* (Hu et al., 2020) | $20 \times c$ | $30 \times c$ | Rest |
| **Graph Regression** | *QM9* (Wu et al., 2018) | 50% | 25% | 25% |
| | *ZINC (Subset)* (Gómez-Bombarelli et al., 2018) | 50% | 25% | 25% |
| **Graph Classification** | *PROTEINS* (Morris et al., 2020) | 50% | 25% | 25% |
| | *AIDS* (Morris et al., 2020) | 50% | 25% | 25% |

# 6 Result and Analysis

## 6.1 Model Performance

**Node Classification:** Table 4 shows the results for node classification accuracy (mean and standard deviation of the top 10 accuracies) with coarsening ratios 0.3 and 0.5 using the **Cluster Nodes** approach. Comparison with other coarsening ratios is in Table 12. Results show that the accuracy is comparable to the baselines. Figure 3 shows the comparison of performance between different experimental setups for different methods of appending nodes on the *Cora* dataset. From the Figure, we observe that **Gc-train-to-Gs-train** offers the best result among the three experimental setups for the node classification task. For the 'None' method of appending nodes, where no additional nodes are added to each subgraph, we notice that it performs poorly compared to other methods, indicating that

Table 3: Results on OGBN-Products. FIT-GNN model uses a coarsening ratio 0.5. **variation_neighborhoods** coarsening algorithm is used. OOM means Out Of Memory.

| | |
|---|---|
| **Full** (Luo et al., 2024) | $0.823 \pm 0.001$ |
| **SGGC** | OOM |
| **GCOND** | OOM |
| **BONSAI** | OOM |
| **FIT-GNN** | $\mathbf{0.894 \pm 0.000}$ |

the Extra Node and Cluster Node methods indeed provide additional information for better empirical performance. We also observe in the 'None' method that, with an increase in the coarsening ratio, the performance drops. This is because, as the coarsening ratio increases, the size of each subgraph gets smaller, hence more edges are cut due to partitioning, leading to more information loss. For the **Gc-train-to-Gs-infer** method, we observe that the performance drops as the coarsening ratio increases. This can be argued because of the change in the distribution of the coarsened graph and the set of subgraphs. When training on $G_s$, the distribution in the train and test sets is similar, hence attaining better performance. While comparing the methods of appending nodes, we infer that the Cluster Node method is performing similarly compared to

Table 4: Results for node classification tasks with accuracy as the metric (higher is better). We use the Cluster Nodes method to append additional nodes to subgraphs and **Gs-train-to-Gs-infer** as experimental setup. **variation_neighborhoods** coarsening algorithm is used.

| Methods | Models | Reduction Ratio (r) | Dataset | | | | |
|---|---|---|---|---|---|---|---|
| | | | *Cora* | *Citeseer* | *Pubmed* | *DBLP* | *Physics* |
| **Full** | GCN | 1.0 | $0.821 \pm 0.002$ | $0.706 \pm 0.002$ | $0.768 \pm 0.020$ | $0.724 \pm 0.002$ | $\mathbf{0.933 \pm 0.000}$ |
| | GAT | 1.0 | $0.809 \pm 0.004$ | $0.700 \pm 0.003$ | $0.740 \pm 0.004$ | $0.713 \pm 0.004$ | $0.909 \pm 0.008$ |
| **SGGC** | GCN | 0.3 | $0.808 \pm 0.003$ | $0.700 \pm 0.001$ | $0.773 \pm 0.002$ | $0.737 \pm 0.011$ | $0.928 \pm 0.003$ |
| | | 0.5 | $0.808 \pm 0.001$ | $0.716 \pm 0.002$ | $\boxed{0.793 \pm 0.002}$ | $0.733 \pm 0.008$ | $0.931 \pm 0.004$ |
| | GAT | 0.3 | $0.686 \pm 0.010$ | $0.635 \pm 0.021$ | $0.773 \pm 0.002$ | $0.626 \pm 0.028$ | $0.870 \pm 0.016$ |
| | | 0.5 | $0.641 \pm 0.026$ | $0.635 \pm 0.024$ | $0.683 \pm 0.024$ | $0.604 \pm 0.047$ | $0.842 \pm 0.018$ |
| **GCOND** | GCN | 0.3 | $0.806 \pm 0.003$ | $0.722 \pm 0.001$ | $0.779 \pm 0.002$ | $0.748 \pm 0.001$ | $0.876 \pm 0.011$ |
| | | 0.5 | $0.808 \pm 0.004$ | $\boxed{\mathbf{0.730 \pm 0.007}}$ | $0.773 \pm 0.004$ | $0.755 \pm 0.002$ | $0.804 \pm 0.005$ |
| | GAT | 0.3 | $0.549 \pm 0.080$ | $0.628 \pm 0.095$ | $0.370 \pm 0.079$ | $0.481 \pm 0.034$ | OOM |
| | | 0.5 | $0.712 \pm 0.059$ | $0.501 \pm 0.085$ | $0.409 \pm 0.044$ | $0.464 \pm 0.030$ | OOM |
| **BONSAI** | GCN | 0.3 | $0.722 \pm 0.011$ | $0.585 \pm 0.003$ | $0.538 \pm 0.103$ | $0.683 \pm 0.039$ | $0.798 \pm 0.007$ |
| | | 0.5 | $0.701 \pm 0.012$ | $0.666 \pm 0.001$ | $0.670 \pm 0.084$ | $0.702 \pm 0.049$ | $0.832 \pm 0.016$ |
| | GAT | 0.3 | $0.680 \pm 0.014$ | $0.566 \pm 0.003$ | $0.540 \pm 0.096$ | $0.665 \pm 0.027$ | $0.791 \pm 0.010$ |
| | | 0.5 | $0.683 \pm 0.006$ | $0.579 \pm 0.004$ | $0.523 \pm 0.133$ | $0.681 \pm 0.025$ | $0.812 \pm 0.016$ |
| **FIT-GNN** | GCN | 0.3 | $0.800 \pm 0.003$ | $0.679 \pm 0.003$ | $0.754 \pm 0.001$ | $\boxed{\mathbf{0.786 \pm 0.001}}$ | $\boxed{0.932 \pm 0.000}$ |
| | | 0.5 | $\boxed{\mathbf{0.829 \pm 0.002}}$ | $0.668 \pm 0.003$ | $0.761 \pm 0.004$ | $0.700 \pm 0.001$ | $0.926 \pm 0.000$ |
| | GAT | 0.3 | $0.761 \pm 0.004$ | $0.655 \pm 0.005$ | $0.754 \pm 0.002$ | $0.771 \pm 0.002$ | $0.885 \pm 0.004$ |
| | | 0.5 | $0.792 \pm 0.003$ | $0.669 \pm 0.004$ | $0.760 \pm 0.003$ | $0.720 \pm 0.002$ | $0.885 \pm 0.003$ |

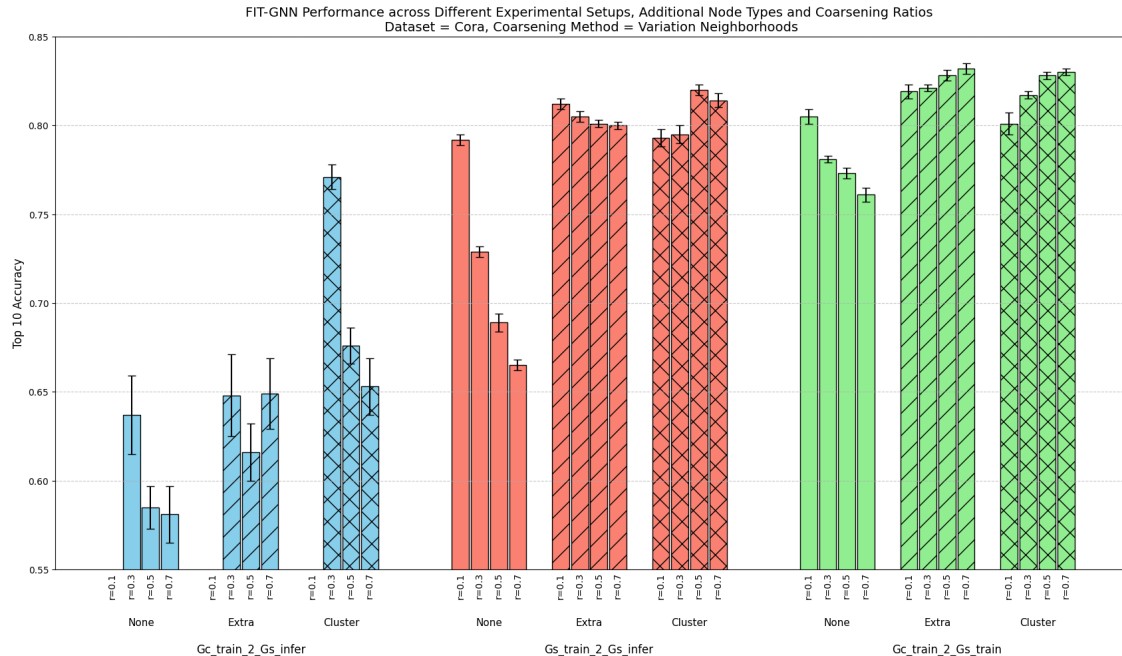

Figure 3: The plot shows the ablation study conducted on the *Cora* dataset to determine which experimental setup performs better than the others. The plot also compares different methods of appending nodes to the subgraphs and how performance changes with varying coarsening ratios. **variation_neighborhoods** coarsening algorithm is used.

the Extra Nodes method, if not better. Table 3 shows results on the *OGBN-Products* dataset. We use the result by Luo et al. (2024) for the full graph setting, which used the OGBN's standard split, while the split used in our work is detailed in Table 2.

Table 5: This table shows the normalized MAE loss (lower is better) for node regression task on different datasets with **Gs-train-to-Gs-infer** experiment setup and **Cluster Nodes** method. **variation_neighborhoods** coarsening algorithm is used.

| Methods | Models | Reduction Ratio (r) | Dataset | | |
| --- | --- | --- | --- | --- | --- |
| | | | *Chameleon* | *Crocodile* | *Squirrel* |
| **Full** | GCN | 1.0 | $0.852 \pm 0.001$ | $0.854 \pm 0.000$ | $0.802 \pm 0.000$ |
| | GAT | 1.0 | $0.846 \pm 0.002$ | $0.850 \pm 0.000$ | $0.805 \pm 0.001$ |
| | SAGE | 1.0 | $0.848 \pm 0.002$ | $0.850 \pm 0.000$ | $0.804 \pm 0.000$ |
| | GIN | 1.0 | $0.843 \pm 0.000$ | $0.852 \pm 0.000$ | $0.796 \pm 0.000$ |
| **FIT-GNN** | GCN | 0.1 | $0.496 \pm 0.005$ | $0.364 \pm 0.001$ | $0.663 \pm 0.003$ |
| | | 0.3 | $0.531 \pm 0.006$ | $0.369 \pm 0.002$ | $0.682 \pm 0.002$ |
| | | 0.5 | $0.531 \pm 0.004$ | $0.371 \pm 0.001$ | $0.722 \pm 0.002$ |
| | | 0.7 | $0.536 \pm 0.009$ | $0.368 \pm 0.003$ | $0.696 \pm 0.002$ |
| | GAT | 0.1 | $0.582 \pm 0.029$ | $0.395 \pm 0.022$ | $0.734 \pm 0.019$ |
| | | 0.3 | $0.537 \pm 0.021$ | $0.383 \pm 0.020$ | $0.724 \pm 0.020$ |
| | | 0.5 | $0.556 \pm 0.018$ | $0.386 \pm 0.018$ | $0.728 \pm 0.023$ |
| | | 0.7 | $0.576 \pm 0.018$ | $0.395 \pm 0.012$ | $0.746 \pm 0.020$ |
| | SAGE | 0.1 | $0.506 \pm 0.006$ | $\mathbf{0.309 \pm 0.014}$ | $\mathbf{0.589 \pm 0.003}$ |
| | | 0.3 | $0.489 \pm 0.003$ | $0.315 \pm 0.001$ | $0.602 \pm 0.003$ |
| | | 0.5 | $\mathbf{0.484 \pm 0.004}$ | $0.317 \pm 0.001$ | $0.618 \pm 0.002$ |
| | | 0.7 | $0.539 \pm 0.009$ | $0.323 \pm 0.000$ | $0.621 \pm 0.004$ |
| | GIN | 0.1 | $0.769 \pm 0.101$ | $0.722 \pm 0.098$ | $0.795 \pm 0.018$ |
| | | 0.3 | $0.808 \pm 0.056$ | $0.697 \pm 0.122$ | $0.796 \pm 0.004$ |
| | | 0.5 | $0.810 \pm 0.062$ | $0.730 \pm 0.091$ | $0.786 \pm 0.032$ |
| | | 0.7 | $0.792 \pm 0.089$ | $0.779 \pm 0.102$ | $0.791 \pm 0.009$ |

**Node Regression:** Table 5 presents the node regression error (mean and standard deviation of the lowest 10 MAE). Interestingly, our results demonstrate that utilizing localized subgraphs for inference (FIT-GNN) significantly improves performance compared to full-graph inference. To investigate this counterintuitive performance leap, we conducted a rigorous ablation study (Appendix G). Our analysis indicates that this improvement is driven by the structural properties of the input data during inference rather than the training regime itself. First, partitioning the graph creates localized contexts that are statistically more homogeneous; we observed that the label standard deviation within individual subgraphs is drastically lower than the global variation, presenting a simpler optimization landscape for the model. Second, while coarsening (e.g., at a ratio of $r = 0.5$) causes a vast majority of nodes to lose a significant portion of their distant 2nd-hop neighborhood, this structural loss actually acts as *implicit adversarial pruning*. In these specific heterophilic graphs, long-range information often introduces noise or adversarial signals. By filtering out this noise through coarsening, the model is able to fully exploit the low-variance local structures, ultimately leading to the observed reduction in regression error.

**Graph Regression:** Table 6 shows the results for the graph regression task on *ZINC (Subset)* and *QM9* dataset. All the results show that the FIT-GNN model's performance is better than the baselines. It is also observed that a lower coarsening ratio yields better loss, which implies that the model performs better on molecular graphs when the subgraph size is large. We observe that finer subgraphs lose out on global information about the molecule, which is necessary for the prediction. Similarly, the performance degrades in the **Gc-train-to-Gc-infer** setup, where we infer on the coarsened graphs. This is because, when a graph is reduced to a smaller graph, the individual node information is lost, which is crucial since, in graphs representing different molecules, components formed by different atoms result in different properties of the molecule.

**Graph Classification:** Table 7 shows graph classification results on *AIDS* and *PROTEINS* datasets, where training and inference are done on $G'$. For baselines like DOSCOND and KIDD, we use 'Graph per class', whereas we use 'Reduction Ratio (r)' as a metric to denote how these algorithms reduce the size of training data. DOSCOND and KIDD propose to generate synthetic graphs that mimic the training data distribution.

Table 6: Results on Graph Regression with MAE as metric (lower is better); We use Extra node and **Gs-train-to-Gs-infer** as experimental setup. **variation_neighborhoods** coarsening algorithm is used.

| Methods | Models | Reduction Ratio (r) | Dataset | | | | |
|---------|--------|---------------------|---------------|----------|----------------|-------------|-------------------------|
| | | | ZINC (Subset) | QM9 ($\mu$) | QM9 ($\Delta\epsilon$) | QM9 (ZPVE) | QM9 ($U^{\text{ATOM}}$) |
| **Full** | GCN | 1.0 | 0.743 | 0.855 | 1.012 | 1.102 | 1.073 |
| | GAT | 1.0 | 0.736 | 0.919 | 1.015 | 1.104 | 1.066 |
| | SAGE | 1.0 | 0.685 | 0.885 | 1.015 | 1.109 | 1.076 |
| | GIN | 1.0 | 0.748 | 0.925 | 1.053 | 1.126 | 1.078 |
| **FIT-GNN** | GCN | 0.1 | 0.625 | 0.713 | 1.223 | **0.447** | 1.223 |
| | | 0.3 | **0.573** | 0.708 | 0.975 | 0.473 | 1.220 |
| | | 0.5 | 0.645 | 0.702 | 1.153 | 0.485 | 1.179 |
| | | 0.7 | 0.651 | 0.715 | 1.189 | 0.456 | 1.240 |
| | GAT | 0.1 | 0.574 | 0.705 | 0.940 | 0.464 | 0.524 |
| | | 0.3 | 0.652 | 0.708 | 1.056 | 0.528 | 0.564 |
| | | 0.5 | 0.620 | 0.693 | 1.227 | 0.475 | 0.561 |
| | | 0.7 | 0.658 | **0.686** | 1.030 | 0.491 | 0.703 |
| | SAGE | 0.1 | 0.735 | 0.726 | **0.739** | 0.526 | 0.905 |
| | | 0.3 | 0.671 | 0.748 | 0.711 | 0.602 | **0.518** |
| | | 0.5 | 0.695 | 0.739 | 0.747 | 0.606 | 0.833 |
| | | 0.7 | 0.645 | 0.722 | 0.677 | 0.659 | 1.005 |
| | GIN | 0.1 | 0.644 | 0.726 | 0.746 | 0.494 | 0.856 |
| | | 0.3 | 0.669 | 0.723 | 0.761 | 0.509 | 0.854 |
| | | 0.5 | 0.631 | 0.725 | 0.848 | 0.737 | 1.058 |

Therefore, to cover all the graphs in the dataset appropriately, this approach generates {1, 10, 50} graphs per class. However, the graph size is kept similar to the graph size in the training set. These approaches also don't cover the test set. Compared to these baselines, our approach transforms each graph into a set of subgraphs and a coarsened graph based on a reduction ratio. As a result, each graph in the training set and test set gets reduced. From our results, we see that FIT-GNN outperforms all the baselines.

Table 14 and 15 present the ablation study on various coarsening algorithms for all the aforementioned tasks. From the results, we observe that **variation_neighborhoods** coarsening algorithm yields consistently better results compared to others.

Table 7: Results on Graph Classification with accuracy as metric (higher is better). We use the extra nodes method and **Gc-train-to-Gc-infer** as experimental setup. **algebraic_JC** coarsening algorithm is used.

| Methods | Models | Graph per class | Dataset | |
|---------|--------|-----------------|---------|----------|
| | | | AIDS | PROTEINS |
| **DOSCOND** | GCN | 1 | 0.785 | 0.590 |
| | | 10 | 0.666 | 0.647 |
| | | 50 | 0.518 | 0.656 |
| | GAT | 1 | 0.608 | 0.652 |
| | | 10 | 0.723 | 0.667 |
| | | 50 | 0.729 | 0.657 |
| **KIDD** | GCN | 1 | 0.399 | 0.416 |
| | | 10 | 0.236 | 0.659 |
| | | 50 | 0.252 | 0.671 |
| | GAT | 1 | 0.441 | 0.416 |
| | | 10 | 0.225 | 0.665 |
| | | 50 | 0.218 | 0.671 |
| | SAGE | 1 | 0.653 | 0.416 |
| | | 10 | 0.361 | 0.650 |
| | | 50 | 0.283 | 0.673 |
| | GIN | 1 | 0.759 | 0.412 |
| | | 10 | 0.637 | 0.655 |
| | | 50 | 0.634 | 0.674 |

| Methods | Models | Reduction Ratio (r) | Dataset | |
|---------|--------|---------------------|---------|----------|
| | | | AIDS | PROTEINS |
| **Full** | GCN | 1.0 | 0.788 | 0.710 |
| | GAT | 1.0 | 0.802 | 0.645 |
| | SAGE | 1.0 | 0.770 | 0.613 |
| | GIN | 1.0 | 0.800 | 0.774 |
| **FIT-GNN** | GCN | 0.1 | 0.810 | 0.783 |
| | | 0.3 | **0.844** | **0.826** |
| | | 0.5 | 0.836 | 0.696 |
| | | 0.7 | 0.793 | 0.783 |
| | GAT | 0.1 | 0.784 | 0.652 |
| | | 0.3 | 0.793 | 0.522 |
| | | 0.5 | 0.810 | 0.739 |
| | | 0.7 | 0.759 | 0.522 |
| | SAGE | 0.1 | 0.793 | 0.696 |
| | | 0.3 | 0.828 | 0.696 |
| | | 0.5 | 0.793 | 0.696 |
| | | 0.7 | 0.819 | **0.826** |
| | GIN | 0.1 | 0.819 | 0.652 |
| | | 0.3 | 0.828 | 0.783 |
| | | 0.5 | 0.793 | 0.478 |
| | | 0.7 | 0.819 | 0.739 |

## 6.2 Inference Time and Memory

As mentioned in Section 4.3, our inference time is less than the standard GNN if certain conditions are met. In Table 8a and 8b, we club all baseline models into one, since all **baselines are inferred on the whole graph**. We empirically show the reduction of time and space by our FIT-GNN model during inference. For recording the inference time, we use the Python **time** package to calculate the difference in time before and after the inference step. In Table 8a, we show the average time to predict for 1000 queries (Prediction for

Table 8: Inference time (sec) (Lower is better) comparison for Node-Level tasks and Graph-Level tasks. We show single-node prediction time using two different coarsening ratios for Node-Level tasks. Here, we used **Cluster Nodes**. We used **Gc-train-to-Gc-infer** experimental setup for the graph-level tasks.

(a) Node-Level Tasks

| Dataset | Baselines | FIT-GNN | |
| --- | --- | --- | --- |
| | | $r = 0.1$ | $r = 0.3$ |
| *Chameleon* | 0.0027 | 0.0016 | **0.0014** |
| *Squirrel* | 0.0081 | 0.0017 | **0.0014** |
| *Crocodile* | 0.0070 | 0.0015 | **0.0015** |
| *Cora* | 0.0026 | **0.0019** | 0.0020 |
| *Citeseer* | 0.0031 | **0.0018** | 0.0019 |
| *Pubmed* | 0.0042 | 0.0019 | **0.0018** |
| *DBLP* | 0.0063 | 0.0020 | **0.0018** |
| *Physics Coauthor* | 0.0252 | 0.0020 | **0.0017** |
| *OGBN-Products* | 0.1762 | 0.0017 | **0.0016** |

(b) Graph-Level Tasks

| Dataset | Baselines | FIT-GNN | |
| --- | --- | --- | --- |
| | | $r = 0.3$ | $r = 0.5$ |
| *ZINC (subset)* | **0.00184** | **0.00184** | 0.00190 |
| *QM9* | **0.00173** | 0.00180 | 0.00191 |
| *AIDS* | 0.00163 | **0.00155** | 0.00163 |
| *PROTEINS* | 0.00165 | **0.00160** | 0.00163 |

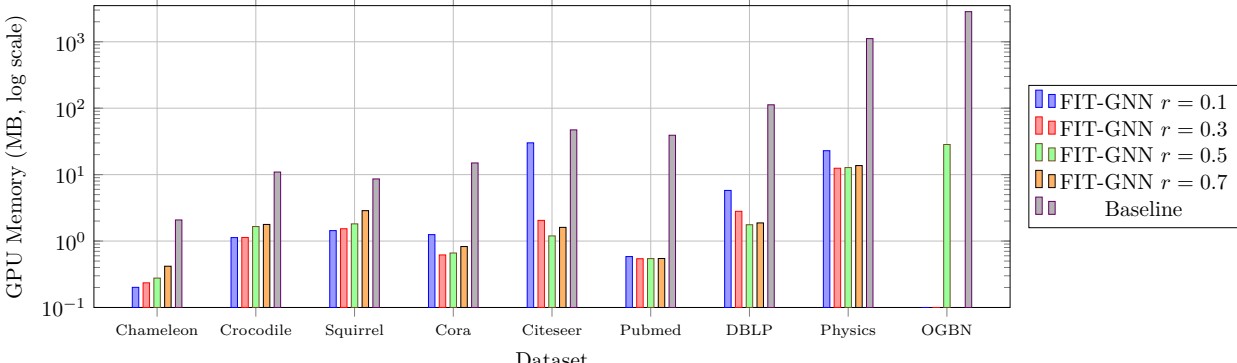

Figure 4: GPU memory consumption in MegaBytes (MB) (log scale) for FIT-GNN (Cluster Node) at different reduction ratios $r$ and Baseline, during inference. **variation_neighborhoods** coarsening algorithm is used.

Node $v_i$) in the Baseline and FIT-GNN models for node regression and classification tasks. The baseline model processes the entire graph, increasing inference time, especially for large graphs. In contrast, the FIT-GNN model only requires the relevant subgraph, resulting in faster predictions. For larger datasets like *OGBN-Products*, baseline inference is not feasible with our computational capacity. Therefore, to show a comparison, we take a subset of *OGBN-Products* with 165000 nodes and 4340428 edges with respect to which we see up to **100× speedup in inference time**. Section C elaborates more on the inference time for datasets with a larger number of nodes. Traditional methods fail to infer for very large datasets due to memory constraints. However, our approach enables inference on the whole graph. Figure 4 and Table 13 show the detailed comparison of memory consumption for different datasets for both Extra Nodes and Cluster Nodes methods, along with the baseline memory, which highlights that the FIT-GNN model uses up to **100× less memory** than the Baseline. Figure 6 shows the time taken to coarsen for different coarsening ratios for the *Cora* dataset. As the coarsening ratio increases, the time also increases, since the number of subgraphs increases.

Table 8b compares the inference time of graph classification and graph regression tasks. We predict for randomly selected 1000 graphs from the test split. The table shows how our method is comparable and sometimes faster than the Baseline. Also, we observe that the inference time increases with a higher coarsening ratio. Because with a higher coarsening ratio, the number of nodes in $G'$ increases, resulting in more edges. Overall, the inference time and memory for all the tasks mentioned are drastically less than the baselines while maintaining the performance.

## 7 Conclusion

In this paper, we have focused on inference time and memory and presented a new way to utilize existing graph coarsening algorithms for GNNs. We have provided theoretical insight corresponding to the number of nodes in the graph for which the FIT-GNN model reduces the time and space complexity. Empirically, we have shown that our method is comparable to the uncoarsened Baseline while being orders of magnitude faster in terms of inference time and consuming a fraction of the memory. Future directions include extending the FIT-GNN methodology to directed, weighted, heterogeneous, and temporal graphs, alongside tackling data-intensive real-world challenges like weather forecasting. Theoretically, we aim to quantify the information loss for the Cluster Node method, investigate its connection to Extra Nodes, and explore novel strategies beyond both to further mitigate graph partitioning loss.

## 8 Acknowledgment

Anirban Dasgupta acknowledges the support by SERB MATRICS and SERB CRG grants and the support from N Rama Rao Chair Professorship.

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

# Appendix

## A   More on Extra Nodes and Cluster Nodes

### A.1   Extra Nodes

**Lemma 4.1.** *Models with* 1 *layer of GNN cannot distinguish between $G$ and $\mathcal{G}_s$ when **Extra Nodes** method is used.*

*Proof.* There are 2 sets of nodes in $G_i$ as follows:

- The set $S_1$ of nodes with 1-hop neighbours in $G_i$.

- The set $S_2$ of nodes with not all 1-hop neighbors in $G_i$

Let $\mathcal{I}_i^1$ be the number of nodes whose information does not get passed on after 1 layer of GNN for $G_i$.

$$\mathcal{I}_i^1 = \left| \bigcup_{v \in S_2} \mathcal{N}_1(v) - V(G_i) \right|$$

$$\text{Now, } \left| \bigcup_{v \in S_1} \{u \in V : u \in \mathcal{N}_1(v) \wedge u \notin G_i\} \right| = 0$$

$$\text{Also, } \mathcal{E}_{G_i} = \bigcup_{v \in S_2} \{u \in V : u \in \mathcal{N}_1(v) \wedge u \notin G_i\}$$

$$\Rightarrow |\mathcal{E}_{G_i}| = \mathcal{I}_i^1$$

Hence, when **Extra Nodes** is used, 1 Layer GNN model cannot distinguish $G$ and $\mathcal{G}_s$. □

To understand the information loss when taking 2 Layers of GNN, we divide $G_i$ into 3 sets of nodes.

- $S_1$: Nodes with 1-hop and 2-hop neighbours in $G_i$.

- $S_2$: Nodes with 1-hop neighbours in $G_i$ but $\exists v$ in 2-hop neighbourhood that is not in $G_i$.

- $S_3$: Node where $\exists v$ in 1-hop and 2-hop neighbourhood that is not in $G_i$.

Let $\mathcal{I}_i^2$ be the number of nodes whose information doesn't get passed on after 2 layer of GNN for $G_i$.

$$\mathcal{I}_i^2 = \left| \bigcup_{v \in S_3} \mathcal{N}_2(v) - V(G_i) \right|$$

When we use **Extra Nodes**, the information loss can be written as follows:

$$\mathcal{I}_i^2 = \left| \bigcup_{v \in S_3} \mathcal{N}_2(v) - V(G_i) - \mathcal{E}_{G_i} \right|$$

The above entity will depend on the density of the subgraphs formed and the number of connections each subgraph shares with each other. An algorithm with an objective to reduce this entity for all subgraphs will lose the least amount of information when **Extra Nodes** method is used.

## A.2 Cluster Nodes

Given a partition matrix $P$, the features of the coarsened node are $X' = P^\top X$. Given a normalized partition matrix, the features of a node $v'_i \in G'$ is the degree-weighted average of the features of nodes in $C_i$. This is one of the functions $f$ used to create the features of the cluster node from $C_i$.

Previously, according to Lemma 4.1, there is no information loss when using the 1 layer of GNN and **Extra Nodes** method. It was also easy to quantify in terms of the number of nodes. However, it is different for **Cluster Nodes**. Let us discuss the issues first.

- Only a weighted version of node information is shared with the subgraph. Suppose $v_c, v_d \in G_i$ is connected to $v_a, v_b \in G_j$. Then the information contributed by these nodes is $\frac{d_a x_a + d_b x_b}{\sum_p d_p}$. Here $d_p$ represents the degree of node $v_p$, and $x_p$ represents the feature of node $v_p$.

- Other node information will also be shared which is $\frac{\sum_{p \neq v_a, v_b} d_p x_p}{\sum_p d_p}$. This will capture further dependencies.

The performance of **Cluster Node** will depend on some distance or similarity metric between $x_c, x_d$ and $f(C_j)$.

# B  More Algorithms

---

**Algorithm 3** Training GNN (Train on $G'$) (Huang et al., 2021)

---

**Require:** $G = (V, E, X)$; Labels $Y$; Node-Model $M$; Loss $\ell$; Number of Layers $L$
1: Apply a coarsening algorithm on $G$, and output a normalized partition matrix $P$;
2: Construct Coarsened graph $G'$ using $P$;
3: Construct feature matrix for $G'$ by $X' = P^\top X$;
4: Construct labels for $G'$ by $Y' = \arg\max(P^\top Y)$ ;
5: $O = M(L, A', D', X')$;                                      ▷ Algorithm 4
6: $Loss = \ell(O, Y')$;
7: Train $M$ to minimize $Loss$;

---

**Algorithm 4** Node-Model$(L, A, D, X)$

---

**Require:** Number of Layers $L$; $A$; $D$; $X^{(0)} = X$;
1: **for** $i = 1$ to $L$ **do**
2:     $X^{(i)} = \sigma(\tilde{D}^{-\frac{1}{2}} \tilde{A} \tilde{D}^{-\frac{1}{2}} X^{(i-1)} \mathcal{W}^{(i-1)})$                       ▷ Equation 1
3: **end for**
4: $Z = X^{(L)} \mathcal{W}^{(L)}$
5: **return** $Z$

---

**Algorithm 5** Graph-Model-$G'(L, A', D', X')$

---

**Require:** Number of Layers $L$; $A'$; $D'$; $X^{(0)} = X'$
1: **for** $i = 1$ to $L$ **do**
2:     $X^{(i)} = \sigma(\tilde{D}'^{-\frac{1}{2}} \tilde{A}' \tilde{D}'^{-\frac{1}{2}} X^{(i-1)} \mathcal{W}^{(i-1)})$                   ▷ Equation 1
3: **end for**
4: $\bar{X} = \text{MaxPooling}(X^{(L)})$
5: $Z = \bar{X} \mathcal{W}^{(L)}$
6: **return** $Z$

---

# C  More on Time and Space complexity

## C.1  Comparison with Baseline

In Lemma 4.2, we provide the conditions that need to be satisfied for better space and time complexities.

For node-level tasks, to obtain lower FIT-GNN inference time complexity as compared to classical GNN, we want the following as per asymptotic analysis,

1. Baseline vs. Single-Node Inference:

$$\underbrace{n^2 d + nd^2}_{\text{Baseline Complexity}} \geq \underbrace{\max_i^k [\bar{n}_i^2 d + \bar{n}_i d^2]}_{\text{FIT-GNN Complexity}} \tag{4}$$

2. Baseline vs. Full-Graph Inference:

$$\underbrace{n^2 d + nd^2}_{\text{Baseline Complexity}} \geq \underbrace{\sum_{i=1}^{k} \left[ (n_i + \phi_i)^2 d + (n_i + \phi_i)d^2 \right]}_{\text{FIT-GNN Complexity}} \tag{5}$$

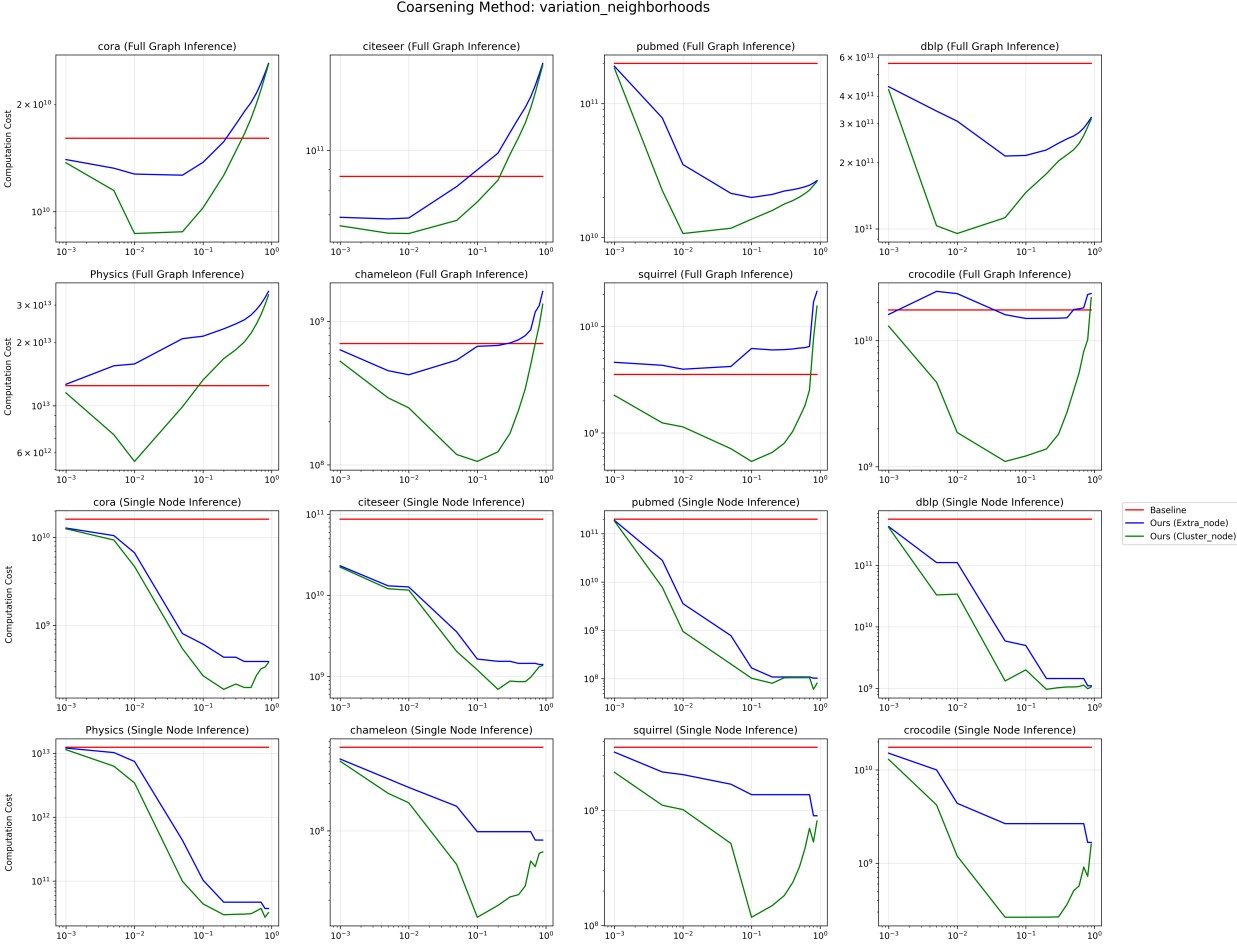

Figure 5: The figure shows the feasibility of our inference methods for different node-level datasets against coarsening ratios. Two important inference setups for each dataset: single-node inference with the computational cost $\mathcal{O}\left(\max_i(\bar{n}_i^2 d + \bar{n}_i d^2) + n\right)$, and full-graph inference with computational cost $\mathcal{O}\left(\sum_{i=1}^{k}(\bar{n}_i^2 d + \bar{n}_i d^2)\right)$, are compared against the baseline (classical GNN) computational cost of $\mathcal{O}\left(n^2 d + nd^2\right)$. The coarsening method used is variation_neighborhoods. Both axes are in logarithmic scale (base 10).

Figure 5 demonstrates the feasibility of our FIT-GNN method for different coarsening ratios with regards to full-graph inference time complexity. We empirically compute the LHS (baseline is classical GNN) and RHS of the Inequality 4 and 5 for various coarsening ratios and plot the results for multiple datasets. The coarsening algorithm used is variation_neighborhoods.

We make the following inferences:

1. There is a clear dataset-dependent computational-cost trade-off across different coarsening ratios. Moreover, the minimum inference time for our approach occurs at different coarsening ratios for different datasets.

2. Under single-node inference setup, FIT-GNN demonstrates increasingly superior inference time performance as compared to classical GNN with increasing coarsening ratio. This is because the individual subgraph sizes decrease with increasing coarsening ratio.

3. We observe lower inference time complexity of Cluster Node method as compared to Extra Node method under both inference setups. This is expected as $|C_{G_i}| \leq |\mathcal{E}_{G_i}|, \ \forall i \in \{1, 2, \ldots, k\}$.

## C.2   Preprocessing Time Complexity

A comparative analysis of our preprocessing (coarsening) overhead against other state-of-the-art scaling methods has been conducted. As shown in the Table 9 below, our preprocessing time complexity is comparable to, and often faster than, existing SOTA methods. Thus, even when end-to-end latency is considered, FIT-GNN maintains its efficiency advantage.

Table 9: Asymptotic time complexities for different methods. $N$ is the number of nodes in the original graph, $M$ is the number of edges and $C$ is the number of classes; rest of the symbols follow notation defined in Section 3 and 4. Blue and Red indicate CPU and GPU respectively - the hardware used for computation. Full Graph setup indicates testing on all the test nodes in a graph while Single Node setup represents testing for a single test node.

| Method | Preprocessing | Training | Inference (Full Graph) | Inference (Single Node) | Model-Agnostic |
|---|---|---|---|---|---|
| **SGGC** | $M + N$ | $k^2d + kd^2$ | $N^2d + Nd^2$ | $N^2d + Nd^2$ | ✓ |
| **GCOND** | $C(N^2 + k^2)d$ $+ C(N + k)d^2$ | $k^2d + kd^2$ | $N^2d + Nd^2$ | $N^2d + Nd^2$ | ✗ |
| **BONSAI** | $M + N$ | $k^2d + kd^2$ | $N^2d + Nd^2$ | $N^2d + Nd^2$ | ✓ |
| **FIT-GNN** | $M + N$ | $kd^2 + k^2d$ $+ \sum_{i=1}^{k}[\bar{n}_i^2 d + \bar{n}_i d^2]$ | $\sum_{i=1}^{k}[\bar{n}_i^2 d + \bar{n}_i d^2]$ | $\max_{i}^{k}[\bar{n}_i^2 d + \bar{n}_i d^2]$ | ✓ |

We further extend our complexity analysis to the practical scenario, where a new test node $v$ is introduced to the graph $G$. This setting is critical for real-world applications where graphs evolve dynamically. We compare three distinct inference strategies for this new node:

1. **Full Graph**: Construct $G_{new}$ by adding $v$ to the original full graph $G$ and infer on $G_{new}$ using the pre-trained network.

2. **2nd-hop Neighborhood**: Sample only the 2nd-hop neighborhood of $v$ from $G$ and infer using the pre-trained network (assuming a 2-layer architecture).

3. **FIT-GNN Subgraph**: Assign $v$ to a relevant subgraph $G_i$ (e.g., the one containing the majority of its 1st-hop neighborhood) and append the necessary Extra/Cluster nodes. Inference is then performed strictly within the modified subgraph $G_i$.

We present the time complexity comparison for these methods in the Table 10 below. The most significant advantage of the FIT-GNN approach is that inference complexity depends solely on the subgraph size $\bar{n}_i$ ($\bar{n}_i \ll n$). In contrast, the Full Graph approach remains tied to the global parameter $n$ due to the necessity of accessing the full adjacency matrix.

Table 10: Time complexity of different inference strategies when a new node $v$ is added to graph $G$

| Inference Strategy | Preprocessing Complexity | Inference Complexity | Notes |
|---|---|---|---|
| **Full Graph** | $\mathcal{O}(1)$ | $\mathcal{O}\left(n^2 d + n d^2\right)$ | Computationally expensive; requires processing the entire graph for one new node. |
| **2nd-hop Neighborhood** | $\mathcal{O}\left(\|\mathcal{N}_1(v)\|\Delta^2\right)$ | $\mathcal{O}\left(\|\mathcal{N}_2(v)\|^2 d + \|\mathcal{N}_2(v)\| d^2\right)$ | $\|\mathcal{N}_j(v)\|$ is the number of nodes in j-hop neighborhood. $\Delta$ is the maximum degree in a graph. |
| **FIT-GNN Subgraph** | $\mathcal{O}(k)$ | $\mathcal{O}\left(\bar{n}_i^2 d + \bar{n}_i d^2\right)$ | Assuming $v$ has maximum neighbors in $G_i$ which has $n_i$ many nodes. |

## D  Dataset Description

(a) Summary of datasets used for Graph classification

| Dataset | Number of Graphs | Average Nodes | Average Edges | Features | Classes |
|---|---|---|---|---|---|
| *PROTEINS* | 1113 | 19 | 72 | 3 | 2 |
| *AIDS* | 2000 | 7 | 16 | 38 | 2 |

(b) Summary of datasets used for Graph regression

| Dataset | Number of Graphs | Average Nodes | Average Edges | Features | Number of Targets |
|---|---|---|---|---|---|
| *QM9* | 130831 | 8 | 18 | 11 | 19 |
| *ZINC (subset)* | 10000 | 11 | 25 | 1 | 1 |

(c) Summary of datasets used for Node classification

| Dataset | Nodes | Edges | Features | Classes |
|---|---|---|---|---|
| *Cora* | 2708 | 5278 | 1433 | 7 |
| *Citeseer* | 3327 | 4552 | 3703 | 6 |
| *Pubmed* | 19717 | 44324 | 500 | 3 |
| *DBLP* | 17716 | 52867 | 1639 | 4 |
| *Physics Coauthor* | 34493 | 247962 | 8415 | 5 |
| *OGBN-Products* | 2449029 | 61859140 | 100 | 47 |

(d) Summary of datasets used for Node regression

| Dataset | Nodes | Edges | Features | Number of Targets |
|---|---|---|---|---|
| *Chameleon* | 2277 | 31396 | 128 | 1 |
| *Squirrel* | 5201 | 198423 | 128 | 1 |
| *Crocodile* | 11631 | 170845 | 128 | 1 |

## E  Experiment Details (Parameters and Device Configuration)

For node classification and node regression tasks, we use Adam Optimizer with a learning rate of 0.01 and $L_2$ regularization with 0.0005 weight. We use Adam Optimizer with a learning rate of 0.0001 and $L_2$ regularization with 0.0005 weight for graph-level tasks. We set epochs to 20 and the number of layers of GCN to 2 for training. We set the hidden dimensions for each layer of GCN to 512.

The device configurations are Intel(R) Xeon(R) Gold 5120 CPU @ 2.20GHz, 256GB RAM, NVIDIA A100 40GB GPU. We use Pytorch Geometric to train our models.

## F  Results

Table 12 presents the results on the node classification task with all the coarsening ratios $\{0.1, 0.3, 0.5, 0.7\}$. We did not compare our experiments with methods such as Cluster-GCN(Chiang et al., 2019) because the method focuses on reducing training time by sampling subgraphs for several iterations. During the inference phase, however, it still performs inference on the entire graph. Consequently, its inference performance would be at most comparable to that of a standard GCN.

Figure 6 shows the time comparison for creating the subgraphs for *Cora* Dataset for different coarsening ratios. It is observed that the None method takes the least amount of time, which is intuitive since the subgraphs are not appended with additional nodes. For the Extra Node Method, the time increases since the subgraph size increases as new nodes are appended. For Cluster Node, we observe the maximum time taken since we are not only looking at the neighboring nodes that are lost due to partition, but also are

Table 12: Results for node classification tasks with accuracy as the metric (higher is better). We use the **Cluster Nodes** method to append additional nodes to subgraphs and **Gs-train-to-Gs-infer** as experimental setup. **variation_neighborhoods** coarsening algorithm is used.

| Methods | Models | Reduction Ratio (r) | Dataset | | | | |
|---|---|---|---|---|---|---|---|
| | | | *Cora* | *Citeseer* | *Pubmed* | *DBLP* | *Physics* |
| **Full** | GCN | 1.0 | $0.821 \pm 0.002$ | $0.706 \pm 0.002$ | $0.768 \pm 0.020$ | $0.724 \pm 0.002$ | $0.933 \pm 0.000$ |
| | GAT | 1.0 | $0.809 \pm 0.004$ | $0.700 \pm 0.003$ | $0.740 \pm 0.004$ | $0.713 \pm 0.004$ | $0.909 \pm 0.008$ |
| **SGGC** | GCN | 0.1 | $0.752 \pm 0.002$ | $0.704 \pm 0.003$ | $0.730 \pm 0.004$ | $0.728 \pm 0.019$ | $0.892 \pm 0.010$ |
| | | 0.3 | $0.808 \pm 0.003$ | $0.700 \pm 0.001$ | $0.773 \pm 0.002$ | $0.737 \pm 0.011$ | $0.928 \pm 0.003$ |
| | | 0.5 | $0.808 \pm 0.001$ | $0.716 \pm 0.002$ | **$0.793 \pm 0.002$** | $0.733 \pm 0.008$ | $0.931 \pm 0.004$ |
| | | 0.7 | $0.805 \pm 0.002$ | $0.711 \pm 0.003$ | $0.786 \pm 0.003$ | $0.733 \pm 0.014$ | $0.931 \pm 0.003$ |
| | GAT | 0.1 | $0.630 \pm 0.001$ | $0.605 \pm 0.025$ | $0.689 \pm 0.033$ | $0.606 \pm 0.017$ | $0.854 \pm 0.008$ |
| | | 0.3 | $0.686 \pm 0.010$ | $0.635 \pm 0.021$ | $0.773 \pm 0.002$ | $0.626 \pm 0.028$ | $0.870 \pm 0.016$ |
| | | 0.5 | $0.641 \pm 0.026$ | $0.635 \pm 0.024$ | $0.683 \pm 0.024$ | $0.604 \pm 0.047$ | $0.842 \pm 0.018$ |
| | | 0.7 | $0.643 \pm 0.019$ | $0.570 \pm 0.041$ | $0.733 \pm 0.023$ | $0.613 \pm 0.018$ | $0.851 \pm 0.026$ |
| **GCOND** | GCN | 0.1 | $0.789 \pm 0.003$ | $0.676 \pm 0.020$ | $0.785 \pm 0.003$ | $0.718 \pm 0.007$ | $0.884 \pm 0.008$ |
| | | 0.3 | $0.806 \pm 0.003$ | $0.722 \pm 0.001$ | $0.779 \pm 0.002$ | $0.748 \pm 0.001$ | $0.876 \pm 0.011$ |
| | | 0.5 | $0.808 \pm 0.004$ | **$0.730 \pm 0.007$** | $0.773 \pm 0.004$ | $0.755 \pm 0.002$ | $0.804 \pm 0.005$ |
| | | 0.7 | $0.793 \pm 0.000$ | $0.680 \pm 0.012$ | $0.775 \pm 0.002$ | $0.751 \pm 0.000$ | $0.914 \pm 0.004$ |
| | GAT | 0.1 | $0.657 \pm 0.067$ | $0.553 \pm 0.116$ | $0.511 \pm 0.079$ | $0.475 \pm 0.057$ | OOM |
| | | 0.3 | $0.549 \pm 0.080$ | $0.628 \pm 0.095$ | $0.370 \pm 0.079$ | $0.481 \pm 0.034$ | OOM |
| | | 0.5 | $0.712 \pm 0.059$ | $0.501 \pm 0.085$ | $0.409 \pm 0.044$ | $0.464 \pm 0.030$ | OOM |
| | | 0.7 | $0.706 \pm 0.062$ | $0.453 \pm 0.118$ | $0.416 \pm 0.036$ | $0.462 \pm 0.033$ | OOM |
| **BONSAI** | GCN | 0.1 | $0.729 \pm 0.008$ | $0.576 \pm 0.006$ | $0.600 \pm 0.098$ | $0.699 \pm 0.031$ | $0.822 \pm 0.014$ |
| | | 0.3 | $0.722 \pm 0.011$ | $0.585 \pm 0.003$ | $0.538 \pm 0.103$ | $0.683 \pm 0.039$ | $0.798 \pm 0.007$ |
| | | 0.5 | $0.701 \pm 0.012$ | $0.666 \pm 0.001$ | $0.670 \pm 0.084$ | $0.702 \pm 0.049$ | $0.832 \pm 0.016$ |
| | | 0.7 | $0.705 \pm 0.009$ | $0.582 \pm 0.007$ | $0.474 \pm 0.020$ | $0.694 \pm 0.023$ | $0.844 \pm 0.015$ |
| | GAT | 0.1 | $0.685 \pm 0.004$ | $0.589 \pm 0.002$ | $0.446 \pm 0.014$ | $0.642 \pm 0.015$ | $0.803 \pm 0.005$ |
| | | 0.3 | $0.680 \pm 0.014$ | $0.566 \pm 0.003$ | $0.540 \pm 0.096$ | $0.665 \pm 0.027$ | $0.791 \pm 0.010$ |
| | | 0.5 | $0.683 \pm 0.006$ | $0.579 \pm 0.004$ | $0.523 \pm 0.133$ | $0.681 \pm 0.025$ | $0.812 \pm 0.016$ |
| | | 0.7 | $0.703 \pm 0.010$ | $0.594 \pm 0.002$ | $0.497 \pm 0.028$ | $0.705 \pm 0.047$ | $0.799 \pm 0.010$ |
| **FIT-GNN** | GCN | 0.1 | $0.790 \pm 0.004$ | $0.701 \pm 0.001$ | $0.757 \pm 0.002$ | $0.755 \pm 0.001$ | $0.909 \pm 0.000$ |
| | | 0.3 | $0.800 \pm 0.003$ | $0.679 \pm 0.003$ | $0.754 \pm 0.001$ | **$0.786 \pm 0.001$** | **$0.932 \pm 0.000$** |
| | | 0.5 | **$0.829 \pm 0.002$** | $0.668 \pm 0.003$ | $0.761 \pm 0.004$ | $0.700 \pm 0.001$ | $0.926 \pm 0.000$ |
| | | 0.7 | $0.813 \pm 0.002$ | $0.664 \pm 0.001$ | $0.756 \pm 0.002$ | $0.746 \pm 0.001$ | $0.925 \pm 0.000$ |
| | GAT | 0.1 | $0.773 \pm 0.021$ | $0.694 \pm 0.003$ | $0.737 \pm 0.002$ | $0.729 \pm 0.010$ | $0.887 \pm 0.003$ |
| | | 0.3 | $0.761 \pm 0.004$ | $0.655 \pm 0.005$ | $0.754 \pm 0.002$ | $0.771 \pm 0.002$ | $0.885 \pm 0.004$ |
| | | 0.5 | $0.792 \pm 0.003$ | $0.669 \pm 0.004$ | $0.760 \pm 0.003$ | $0.720 \pm 0.002$ | $0.885 \pm 0.003$ |
| | | 0.7 | $0.797 \pm 0.003$ | $0.662 \pm 0.003$ | $0.759 \pm 0.002$ | $0.739 \pm 0.002$ | $0.903 \pm 0.003$ |

keeping a track of the cluster to which these neighboring nodes belong, corresponding to which, we add the cross-cluster edges.

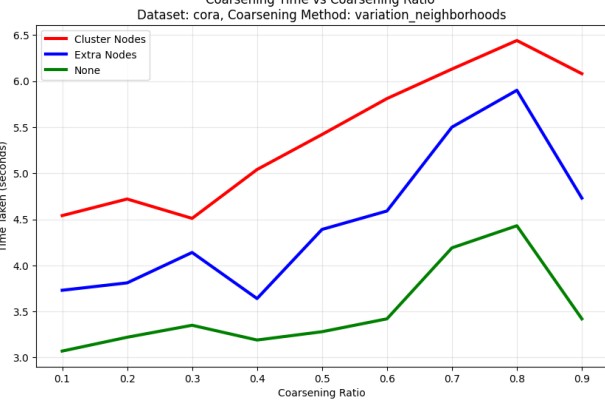

Figure 6: The plot shows how the coarsening time varies for the *Cora* dataset for different coarsening ratios using different methods of appending nodes to the subgraphs. **variation_neighborhoods** coarsening algorithm is used.

Table 13 shows the maximum GPU memory consumption during inference for different datasets with different coarsening ratios and methods of appending nodes. Note that this memory consumption is to store the graph and the weight matrix in the memory only.

Table 13: Summary of maximum GPU memory consumption during inference of datasets used for node-level tasks. All units are in MegaBytes (MB)

| Dataset | Appending Nodes | FIT-GNN | | | | Baseline |
|---|---|---|---|---|---|---|
| | | **r= 0.1** | **r= 0.3** | **r= 0.5** | **r= 0.7** | |
| *Chameleon* | Cluster Node | 0.201 | 0.235 | 0.277 | 0.418 | 2.078 |
| | Extra Node | 0.622 | 0.622 | 0.622 | 0.564 | |
| *Crocodile* | Cluster Node | 1.127 | 1.132 | 1.654 | 1.779 | 10.937 |
| | Extra Node | 4.586 | 4.586 | 4.586 | 4.586 | |
| *Squirrel* | Cluster Node | 1.436 | 1.532 | 1.815 | 2.869 | 8.614 |
| | Extra Node | 7.190 | 7.190 | 7.190 | 7.190 | |
| *Cora* | Cluster Node | 1.249 | 0.618 | 0.661 | 0.827 | 14.992 |
| | Extra Node | 1.906 | 1.103 | 0.970 | 0.970 | |
| *Citeseer* | Cluster Node | 30.097 | 2.048 | 1.195 | 1.607 | 47.170 |
| | Extra Node | 30.097 | 2.490 | 1.835 | 1.835 | |
| *Pubmed* | Cluster Node | 0.584 | 0.542 | 0.544 | 0.546 | 39.166 |
| | Extra Node | 0.778 | 0.562 | 0.562 | 0.562 | |
| *DBLP* | Cluster Node | 5.785 | 2.803 | 1.761 | 1.875 | 112.514 |
| | Extra Node | 28.271 | 6.008 | 2.308 | 2.207 | |
| *Physics Coauthor* | Cluster Node | 22.911 | 12.481 | 12.767 | 13.688 | 1115.079 |
| | Extra Node | 41.066 | 19.936 | 19.936 | 19.936 | |
| *OGBN-Products* | Cluster Node | – | – | 28.399 | – | 2840.706 |

To validate the use of "variation_neighborhoods" as a coarsening algorithm for all the results presented, we show an ablation study for different coarsening algorithms in Table 14 and 15.

Table 14: The table shows the FIT-GNN ablation study to compare various coarsening methods on *Cora* and *Chameleon* datasets, and the metrics for each dataset are accuracy (higher the better) and normalized MAE (lower the better), respectively.

| | *Cora* | | *Chameleon* | |
|---|---|---|---|---|
| **Coarsening Method** | **r = 0.1** | **r = 0.3** | **r = 0.1** | **r = 0.3** |
| **variation_neighborhoods** | $0.790 \pm 0.004$ | $0.800 \pm 0.003$ | **$0.496 \pm 0.005$** | **$0.489 \pm 0.003$** |
| **algebraic_JC** | **$0.827 \pm 0.006$** | **$0.804 \pm 0.003$** | $0.544 \pm 0.005$ | $0.504 \pm 0.003$ |
| **kron** | $0.758 \pm 0.004$ | $0.800 \pm 0.002$ | $0.571 \pm 0.004$ | $0.580 \pm 0.013$ |
| **heavy_edge** | $0.736 \pm 0.012$ | $0.773 \pm 0.006$ | $0.572 \pm 0.002$ | $0.531 \pm 0.002$ |
| **variation_edges** | $0.484 \pm 0.006$ | $0.471 \pm 0.012$ | $0.513 \pm 0.003$ | $0.542 \pm 0.005$ |
| **variation_cliques** | $0.751 \pm 0.012$ | $0.801 \pm 0.009$ | $0.674 \pm 0.009$ | $0.679 \pm 0.008$ |

Table 15: The table shows the FIT-GNN ablation study to compare various coarsening methods on *PROTEINS* and *ZINC (subset)* datasets. The metric for the *PROTEINS* dataset is accuracy (higher is better), and for the *ZINC (subset)* is normalized MAE (lower is better)

| | PROTEINS | | ZINC (subset) | |
|---|---|---|---|---|
| **Coarsening Method** | **r = 0.3** | **r = 0.5** | **r = 0.3** | **r = 0.5** |
| **variation_neighborhoods** | 0.652 | **0.739** | **0.573** | **0.620** |
| **algebraic_JC** | **0.826** | **0.739** | 0.823 | 0.810 |
| **kron** | 0.522 | 0.652 | 0.663 | 1.201 |
| **heavy_edge** | 0.565 | 0.609 | 2.114 | 1.227 |
| **variation_edges** | 0.652 | 0.565 | 1.548 | 1.883 |
| **variation_cliques** | 0.696 | 0.652 | 0.689 | 0.742 |

# G   Detailed Ablation Study on Node Regression Performance

To understand the counterintuitive performance gains observed in node regression tasks when using FIT-GNN, we conducted a comprehensive ablation study. Specifically, we isolated the source of the performance gap to determine why restricting inference to localized subgraphs outperforms full-graph inference.

## G.1   Impact of Inference Input vs. Training Regime

Our first objective was to determine whether the performance gain stemmed from the training methodology itself or the structure of the input data (subgraphs) during inference. We evaluated two distinct setups on the Crocodile dataset using a GCN convolutional layer:

- **Setup A:** Train on subgraphs → Infer on the full graph.

- **Setup B:** Train on the full graph → Infer on the full graph.

Table 16: Comparison of training and inference setups to isolate the source of performance gains.

| Train Setup | Inference Setup | Mean Absolute Error (MAE) |
|---|---|---|
| Full Graph | Full Graph | 0.852 |
| Subgraphs | Full Graph | 0.865 |
| Subgraphs (FIT-GNN) | Subgraphs | **0.364** |

As shown in Table 16, the performance of Setup A and Setup B is nearly identical, indicating that training on subgraphs alone does not account for the drastic MAE reduction. The substantial performance leap in FIT-GNN only occurs when subgraphs are used as the inference input, confirming that the local structural input drives the improvement.

## G.2   Subgraph Optimization Landscape

We further hypothesized that partitioning the graph into subgraphs presents a simpler optimization landscape for the model. To quantify this, we compared the variation of labels locally (within individual subgraphs) versus globally across the entire graph. We computed the Label Standard Deviation for node regression datasets and Label Entropy for node classification datasets.

Table 17: Comparison of global label variation versus average local subgraph variation.

| Dataset | Metric Used | Global Variation | Subgraph Variation (Avg) |
|---|---|---|---|
| **Cora** | Entropy | 1.8311 | 0.1245 |
| **Citeseer** | Entropy | 1.7533 | 0.1572 |
| **Chameleon** | Standard Deviation | 2.1329 | 0.0689 |
| **Squirrel** | Standard Deviation | 1.7639 | 0.1284 |

The results in Table 17 demonstrate that the variation of labels within individual subgraphs is drastically lower than the global variation. By coarsening the graph, we create local contexts that are statistically more homogeneous, allowing the GNN to specialize more effectively and navigate a smoother optimization landscape during inference.

## G.3   Structural Information Loss and Implicit Adversarial Pruning

Finally, we investigated the role of structural information loss. Using a coarsening ratio of $r = 0.5$, we empirically computed the fraction of the 2nd-hop neighborhood lost for each node.

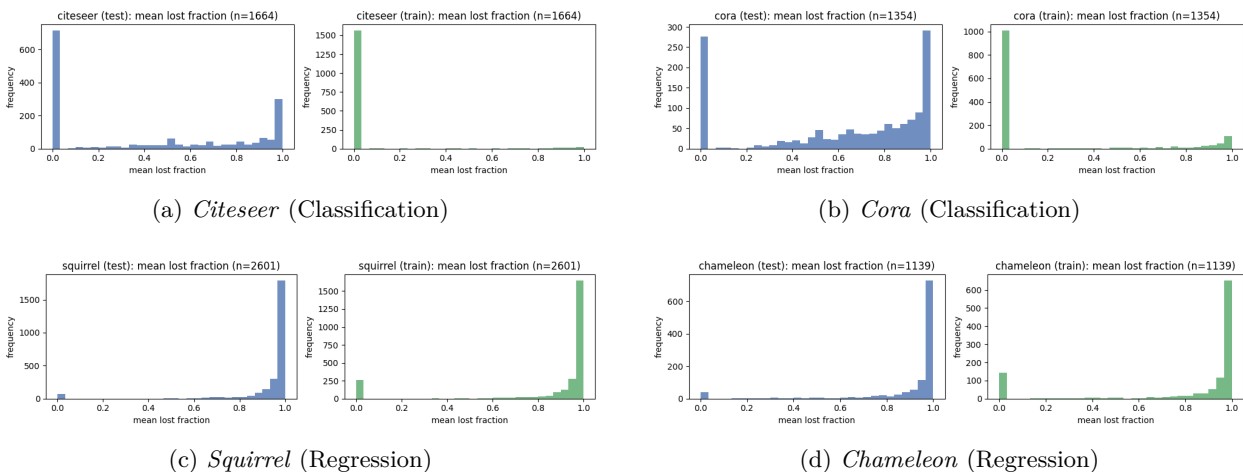

(a) *Citeseer* (Classification)  (b) *Cora* (Classification)

(c) *Squirrel* (Regression)  (d) *Chameleon* (Regression)

Figure 7: Histograms detailing the fraction of the 2nd-hop neighborhood lost for each node at a coarsening ratio of $r = 0.5$. A distinct difference in distribution is visible between classification (a, b) and regression (c, d) datasets.

**Observations and Interpretation:** The histograms in Figure 7 reveal a stark contrast between task types. In the node classification datasets (Cora, Citeseer), a significant number of nodes retain their 2nd-hop neighborhood entirely (lost fraction close to 0). Conversely, in the node regression datasets (Squirrel, Chameleon), the vast majority of nodes lose a highly significant portion of their 2nd-hop neighborhood (lost fraction close to 1).

Given the superior performance of FIT-GNN on these regression tasks, we interpret this structural loss as a form of **implicit adversarial pruning**. In specific heterophilic graphs, long-range information (such as 2nd-hop neighbors) acts as noise or an adversarial signal. The coarsening algorithms utilized by FIT-GNN implicitly filter out this distant noise. This pruning allows the model to fully exploit the low-variance, highly homogeneous local structures identified in Section G, thereby drastically reducing regression error.

