# OpenReview forum: "FIT-GNN: Faster Inference Time for GNNs that ‘FIT’ in Memory Using Coarsening"
_TMLR — Accepted by TMLR_

### Review · Reviewer_UgrG · 2026-01-25

**Summary Of Contributions:**

This paper studies an important problem of scalable graph neural networks. The authors propose a novel approach by reducing computational burden during the inference phase using graph coarsening. Experiments show the effectiveness of the proposals.

**Audience:**

Yes

**Audience Explanation:**

Graph neural networks are widely studied by existing methods in the machine learning area.

**Broader Impact Concerns:**

It would be better to include a subsection to show how this paper can advance the GNN area, and even the machine learning area.

**Claims And Evidence:**

Yes

**Claims Explanation:**

The claims are supported by extensive experiments.

**Requested Changes:**

1. It would be better to include theoretical computational complexity analysis, i.e., space and time complexities.
2. It is suggested to include more experiments regarding efficiency, such as training time, inference time, and FLOPs.
3. The future work seems to be very general. It would be better to provide more specific descriptions.

---

> ### Author Response · Authors · 2026-01-28
> **Response to Requested Changes**
>
> We sincerely thank you for your time and the constructive feedback provided on our submission. We appreciate the opportunity to clarify our contributions and have addressed your specific concerns below.
>
> ---
>
> # 1. Computational Complexity Analysis
>
> > *It would be better to include theoretical computational complexity analysis, i.e., space and time complexities.*
>
> We would like to direct reviewers attention to **Section 4.3 (Time and Space Complexity)**, where we have provided a detailed theoretical analysis of both time and space complexity for our proposed methodology.
>
> * **Comparison with Baselines:** We explicitly compare our method's complexity against standard GNNs and other coarsening baselines (such as SGGC) to demonstrate the theoretical advantages.
> * **Formal Proofs:** We present **Lemma 4.2** and corresponding derivations to formally bound the inference time complexity. This analysis confirms that our approach achieves lower computational costs theoretically, supporting the empirical speedups observed.
>
> ---
>
> # 2. Efficiency Experiments (Training Time, Inference Time, and FLOPs)
>
> > *It is suggested to include more experiments regarding efficiency, such as training time, inference time, and FLOPs.*
>
> We agree that efficiency metrics are central to our paper's contribution. We have included extensive experimental results to substantiate our claims:
>
> * **Inference Time:** **Table 8** provides a comprehensive comparison of inference times across multiple datasets (e.g., Chameleon, Cora) for both node-level and graph-level tasks. The results show our method achieving significant speedups compared to baselines.
>
> * **Memory Consumption:** **Figure 3** illustrates the memory usage for different datasets, highlighting that FIT-GNN significantly reduces memory requirements compared to baselines, enabling inference on large-scale graphs.
>
> * **Coarsening Time:** We have also included **Figure 6** in the Appendix to show the time required for the coarsening process itself.
> * **FLOPs Analysis:** Per your request, we have conducted an additional analysis of FLOPs:
>
> | Dataset | Coarsening Ratio | Method FLOPs | Baseline FLOPs | Reduction Factor |
> | :--- | ---: | ---: | ---: | ---: |
> | **Physics** | 0.1 | 2.27e+08 | 1.58e+11 | 695.9x |
> | | 0.3 | 1.11e+08 | 1.58e+11 | 1423.4x |
> | | 0.5 | 7.88e+07 | 1.58e+11 | 2002.6x |
> | | 0.7 | 6.89e+07 | 1.58e+11 | 2290.7x |
> | **citeseer** | 0.1 | 1.74e+07 | 7.19e+09 | 412.6x |
> | | 0.3 | 1.15e+07 | 7.19e+09 | 626.6x |
> | | 0.5 | 9.23e+06 | 7.19e+09 | 779.2x |
> | | 0.7 | 8.68e+06 | 7.19e+09 | 828.6x |
> | **cora** | 0.1 | 4.46e+07 | 2.71e+09 | 60.6x |
> | | 0.3 | 8.66e+06 | 2.71e+09 | 312.5x |
> | | 0.5 | 6.22e+06 | 2.71e+09 | 435.2x |
> | | 0.7 | 5.71e+06 | 2.71e+09 | 473.6x |
> | **dblp** | 0.1 | 2.33e+07 | 1.95e+10 | 839.4x |
> | | 0.3 | 1.42e+07 | 1.95e+10 | 1375.5x |
> | | 0.5 | 9.25e+06 | 1.95e+10 | 2114.1x |
> | | 0.7 | 8.59e+06 | 1.95e+10 | 2275.7x |
> | **pubmed** | 0.1 | 1.37e+07 | 1.02e+10 | 750.5x |
> | | 0.3 | 6.04e+06 | 1.02e+10 | 1696.5x |
> | | 0.5 | 3.81e+06 | 1.02e+10 | 2692.8x |
> | | 0.7 | 3.23e+06 | 1.02e+10 | 3168.9x |
>
> **Table:** FLOPs comparison across datasets and coarsening ratios (Cluster Node variant) while performing a single-node inference task for 1000 nodes. The reduction factor is computed as (Baseline FLOPs)/(Method FLOPs).
>
> ---
>
> # 3. Specificity of Future Work
>
> > *The future work seems to be very general. It would be better to provide more specific descriptions.*
>
> Thank you for this observation. We refine our future work section to focus on three concrete directions:
>
> * **Quantification of Information Loss:** Unlike the *Extra Node* method, where we have theoretical bounds (Lemma 4.1), the information loss for the *Cluster Node* method is currently not fully quantified. We plan to develop a rigorous theoretical framework to measure this loss.
>
> * **Mitigation Strategies:** We aim to explore novel methods beyond *Extra* and *Cluster* nodes to further mitigate the information loss caused by graph partitioning.
> * **Temporal Extension:** We intend to extend the FIT-GNN methodology to a temporal graph setup, investigating how coarsening can be dynamically adapted for time-evolving networks.

---

> ### Author Response · Authors · 2026-01-28
> **Response to Broader Impact Concerns**
>
> > *It would be better to include a subsection to show how this paper can advance the GNN area, and even the machine learning area.*
>
> We will add a discussion on the broader impact of our work. While substantial research has focused on optimizing the *training* efficiency of GNNs (e.g., via sampling or distillation), the computational burden of *inference* remains a critical bottleneck.
>
> * **Bridging the Gap:** Our paper explicitly highlights this gap. By enabling GNNs to "fit" in memory during inference, we open the door for deploying powerful GNN models on resource-constrained devices (e.g., edge devices) and on extremely large graphs that were previously intractable.
> * **Motivating New Research:** We believe this work will motivate further research into "inference-first" optimizations, shifting the community's focus towards practical deployment challenges in real-world scenarios where latency and memory constraints are paramount.
>
> ---
>
> We hope these responses and the additional data provided adequately address your concerns. We remain open to further feedback to improve the quality of our manuscript.

---

### Review · Reviewer_HnZE · 2026-01-27

**Summary Of Contributions:**

Summary
The paper proposes a framework to make GNN inference computationally efficient, measured in terms of time and memory, by performing inference on a part of the total graph. The idea is to run both training and inference on a set of subgraphs obtained by applying a graph coarsening method to the original graph, rather than using coarsening only to speed up training, as in prior work.
Specifically, as partitioning cuts inter-subgraph edges, the method augments each subgraph with additional context via two existing strategies: Extra Nodes and Cluster Nodes. During training, the method trains a standard GNN by looping over subgraphs and computing the loss only on training nodes in each subgraph. In the experimental section, the paper evaluates multiple training and inference regimes, assesses the methods on node and graph classification and regression tasks across various datasets, and provides a time/memory complexity analysis and a theoretical condition.

Strengths
- The method is simple and straightforward; it essentially uses existing methods to partition the graph, adds boundary context, and then trains and infers on subgraphs. Moreover, two different graph coarsening methods (Extra Nodes, Cluster Nodes) are ablated and discussed.
- The method seems to scale to cases where baselines lead to out-of-memory problems; for example, Table 5 shows that on OGBN-Products, different baseline methods lead to out-of-memory while the proposed approach still runs and achieves good accuracy. The paper also reports clear empirical memory savings; specifically, Table 13 shows large reductions in peak GPU memory during inference.

Weaknesses
- My first major concern is the novelty of the approach. The contribution is incremental, as the core coarsening algorithms (Extra Nodes and Cluster Nodes) have already been proposed before for training GNNs. It seems the new angle is to use these for inference-time scaling and to package them together across tasks.
- Another major concern revolves around clarity, notation, and, in general, the presentation quality of the manuscript. The paper is not always clear; for example, the description of the Extra Nodes and Cluster Nodes methods—including the clarity of Figure 2—is quite brief and unclear. The notation is also not helping improve clarity; for example, in Section 3.1, G is a weighted undirected graph, and W is the edge-weight matrix. However, later in the same section, the notation for an adjacency matrix is used, which indicates a binary, unweighted graph. The notation using an unweighted graph is then used in Section 3.2, equation (1). Also, k=n×r, k∈Z: the round, floor, or ceiling function should be used. Moreover, the writing is dense, explanations are limited, and pointers to Algorithms, Tables, and figures are scattered throughout the paper, making it difficult to understand the training and inference process of the proposed approach.
- Continuing from my previous comment, the theoretical analysis in Section 3.1 is also quite convoluted; for example, it is not clear how the time and space complexities are calculated in Table 1. Moreover, the notation in this section is not clear, and the proof contains no explanations that walk the reader through the proof.
- The paper presents several experiments, but the reasoning and explanations of the observed results are missing. For node regression, the method’s MAE is dramatically lower than the full GNN across multiple model ratios in Table 3. That seems unusual for a method that changes computation locality; it may be correct, but it deserves deeper analysis to rule out confounds (normalization, split differences, leakage assumptions, hyperparameter mismatches).
- It is not clear whether the preprocessing time to coarsen the graph is factored into the complexity analysis; as far as I understand, the evaluation largely focuses on runtime and memory after subgraphs are formed. If this is the case, this preprocessing time should be added to the inference time. In fact, as shown in Figure 6, the coarsening time increases with the coarsening ratio and also the storage of various subgrains, leading to increased complexity.

**Audience:**

No

**Audience Explanation:**

Although the paper addresses an important practical issue, the current manuscript does not present the ideas and evidence with enough clarity and rigor for the findings to be reliably interpreted. The incremental novelty and unclear theoretical and empirical claims might limit the interest of the TMLR audience.

**Broader Impact Concerns:**

I do not have major broader-impact concerns. The work aims at reducing inference time and memory for GNNs, which is generally beneficial.

**Claims And Evidence:**

No

**Claims Explanation:**

While the paper reports promising reductions in inference time and memory, several claims lack sufficiently clear and convincing evidence. In particular, the theoretical analysis and complexity tables are difficult to verify due to unclear notation and an opaque proof, and the empirical results lack the analysis needed to justify findings. Moreover, the reported inference efficiency appears to exclude costs such as coarsening time and subgraph construction overhead, which are directly relevant to the claimed inference-time gains.

The manuscript needs a thorough clean-up so the method is easy to follow; this also includes notation and explanations. In particular, the description of Extra Nodes and Cluster Nodes (and Figure 2) should be expanded and rewritten for readability, and the notation should be made consistent. The text should also be streamlined, reducing density and improving the explanation of the method. Basically, the paper should clearly present the proposed training and inference strategies (against the SoTA) and provide the technical details needed to understand the contribution. On the theoretical analysis, Section 3.1 should be rewritten to make the complexity claims verifiable. The derivation of Table 1 should be explained step by step, with clearly stated assumptions, and the proof (including Lemma 4.2) should include explanatory text that guides the reader through the argument rather than presenting algebra without interpretation.

The paper should include an analysis that explains the key trends and validates that the results are not due to confounds. In particular, the unusually large gains in node regression (Table 3) relative to the full GNN warrant deeper discussion, so the reader can trust that the improvements stem from the proposed inference/training approach.

Finally, in the inference-time claims, it should be made clear whether coarsening time and subgraph construction (and storage) overhead are included; if they are not, they should be accounted for in the reported inference cost.

**Requested Changes:**

The paper should more clearly articulate what is technically new beyond reusing existing coarsening ideas (Extra Nodes / Cluster Nodes) and applying them at inference time. This requires clearer positioning against the state of the art and a stronger statement of the method's technical advantages over prior work.

---

> ### Author Response · Authors · 2026-02-07
> **Response to Reviewer**
>
> We thank the reviewer for their detailed and constructive feedback. We appreciate the time taken to evaluate our work and the insightful comments. Below, we address each concern point by point.
>
> # 1. Clarity, Notation, and Presentation
> > Reviewer Comment: A major concern revolves around clarity, notation, ... to understand the training and inference process of the proposed approach.
>
> We appreciate the reviewer pointing out these areas for improvement. Regarding the *Extra Nodes* and *Cluster Nodes* strategies, we would like to clarify that these concepts are formally defined in Equation 2 and Equation 3  of the manuscript, respectively. Furthermore, Figure 2  was designed to provide a visual representation of how these nodes are appended to subgraphs to mitigate information loss. However, we acknowledge that the textual description accompanying these definitions can be expanded to improve readability, and we will revise the manuscript to ensure these strategies are explained more intuitively.
>
> Regarding the notation in Section 3.1, we accept the oversight. We inadvertently mixed notation for weighted graphs ($W$) and unweighted binary adjacency matrices ($A$). We will modify it to $A \in \mathbb{R}^{n\times n}$ where $A_{ij}$ is the edge weight between nodes $v_i$ and $v_j$. Additionally, we will correct the definition of the number of subgraphs to $k=\lfloor n\times r\rfloor$ to ensure mathematical precision.
>
> # 2. Theoretical Analysis and Complexity
> > Reviewer Comment: Continuing from my previous comment, ... that walk the reader through the proof.
>
> We would like to highlight that Section 3.1 presents the graph notations. Section 4.3 showcases the time and space complexities. We apologize if the derivation of the complexity analysis was not immediately clear. Our complexity analysis is derived directly from the standard Graph Convolutional Network (GCN) computation model described in Equation 1.
>
> To clarify the entries corresponding to time complexities in Table 1:
> - *Classical*: For a graph with $n$ nodes, and feature dimension $d$, a single layer requires matrix multiplcations of $A,X,\mathcal{W}$ of dimension $(n\times n), (n\times d)$ and $(d\times d)$ respectively. These matrix multiplications require $\mathcal{O}(n^2d+nd^2)$.
> - *Coarsening*: Coarsening algorithms reduce the graph with $n$ nodes to a graph with $k$ nodes. As a result, the dimension of the reduced adjacency matrix is $(k\times k)$ and the dimension of the feature matrix becomes $(k\times d)$. Therefore, the training time complexity of this approach is $\mathcal{O}(k^2d+kd^2)$. However, during the inference phase, this approach still requires the full graph as input to the model, hence the time complexity corresponding to that is $\mathcal{O}(n^2d+nd^2)$.
> - *FIT-GNN*: In our approach, the training is done on a coarsened graph with $k$ nodes and a set of $k$ subgraphs. Assuming the $i$-th subgraph has $\bar{n}_i$ nodes (where $\bar{n}_i=n_i+\phi_i$, accounting for original nodes plus appended extra/cluster nodes), the time complexity for that particular subgraph becomes $\bar{n}^2_id+\bar{n}_id^2$. Therefore, the total training time complexity is $\mathcal{O}(k^2d+kd^2+\sum^k _{i=1}[\bar{n}^2_id+\bar{n}_id^2])$. The time complexity for inference on all subgraphs (full-graph inference) is $\mathcal{O}(\sum^k _{i=1}\[\bar{n}^2_id+\bar{n}_id^2\])$.
>
> The space complexity follows a similar logic, comparing the memory required to store the full adjacency and feature matrices versus the maximum memory required to store the largest single subgraph during sequential inference. We will add a step-by-step derivation in the Appendix of the revised paper to make this explicit and verifiable.

---

> ### Author Response · Authors · 2026-02-07
> **Response to Reviewer**
>
> # 3. Analysis of Node Regression Performance
> > Reviewer Comment: The paper presents several experiments, ... leakage assumptions, hyperparameter mismatches).
>
> We agree that this observation is counterintuitive and warrants a deep scientific investigation. To verify that these gains are not spurious, we conducted a rigorous ablation study to isolate the source of the performance gap.
> ## Hypothesis 1:
> We hypothesized that the performance gain stems from the structure of the input data (subgraphs) during inference, rather than the training regime. To test this, we compared two setups:
> 1. *Setup A*: Train on subgraphs $\rightarrow$ Infer on the full graph.
> 2. *Setup B*: Train on the full graph $\rightarrow$ Infer on the full graph.
>
> |Train |Inference|Loss|
> |-|-|-|
> |Full Graph|Full Graph|0.852|
> |Subgraphs|Full Graph|0.865|
> |Subgraphs|Subgraphs|0.364|
>
> *Result*: We experimented on Crocodile dataset with GCN as convolution layer. The performance of Setup A and Setup B was similar. This indicates that the training methodology alone does not account for the drastic MAE reduction. Consequently, the performance leap in FIT-GNN (Train on subgraphs $\rightarrow$ Infer on subgraphs) must be driven by the use of subgraphs as the inference input.
> ## Hypothesis 2:
> We further hypothesized that subgraphs present a "simpler" optimization landscape. To quantify this, we computed the **Label Standard Deviation per subgraph** and compared it to the global Label Standard Deviation for node regression tasks (similarly, **Label Entropy** was used for classification).
>
> *Observation*: As detailed in the table below, the standard deviation of labels within individual subgraphs is drastically lower than the global standard deviation. This suggests that by partitioning the graph, we create local contexts that are statistically more homogeneous, allowing the GNN to specialize more effectively during inference.
>
> |Dataset|Metric Used|Global Variation|Subgraph Variation (Avg)|
> |:---|:---|---:|---:|
> |**Cora**|Entropy|1.8311|0.1245|
> |**Citeseer**|Entropy|1.7533|0.1572|
> |**Chameleon**|Standard Deviation|2.1329|0.0689|
> |**Squirrel**|Standard Deviation|1.7639|0.1284|
> ## Hypothesis 3:
> Finally, we hypothesized that structural information loss plays a role. We empirically computed the **fraction of the 2nd-hop neighborhood lost** for each node due to coarsening at a ratio of $r=0.5$. We request the reviewers to kindly open the anonymous link to look into the plots.
>
> citeseer_lost_frac $\rightarrow$ https://i.ibb.co/tww9PKDh/citeseer-lost-frac.png
>
> cora_lost_frac $\rightarrow$ https://i.ibb.co/PGZMB30K/cora-lost-frac.png
>
> squirrel_lost_frac $\rightarrow$ https://i.ibb.co/rRHKPTR4/squirrel-lost-frac.png
>
> chameleon_lost_frac $\rightarrow$ https://i.ibb.co/fVbxWCD2/chameleon-lost-frac.png
>
> *Observations*:
> - *Node Regression Datasets*: The histogram of neighborhood loss reveals that a vast majority of nodes lose a significant portion of their 2nd-hop neighborhood (Lost fraction close to 1).
> - *Node Classification Datasets*: In contrast, a significant number of nodes retain their 2nd-hop neighborhood entirely (Lost fraction close to 0).
>
> *Interpretation*: We believe the superior performance in node regression is a result of **implicit adversarial pruning**. The high loss of 2nd-hop neighbors in regression datasets suggests that long-range information in these specific heterophilic graphs may act as noise or adversarial signal. FIT-GNN utilises coarsening algorithms which implicitly filters this out, allowing the model to exploit the low-variance local structure identified in the above analysis. This explains why restricting inference to the local subgraph (FIT-GNN) outperforms full-graph inference for these specific tasks.

---

> ### Author Response · Authors · 2026-02-07
> **Response to Reviewer**
>
> # 4. Preprocessing Time Complexity
> > Reviewer Comment: It is not clear whether the preprocessing time ... leading to increased complexity.
>
> We acknowledge this point. In the submitted manuscript, the focus was strictly on the inference-phase latency and memory "on the device."
>
> We have performed a comparative analysis of our preprocessing (coarsening) overhead against other state-of-the-art scaling methods. As shown in the table below, our preprocessing times are comparable to, and often faster than, existing SOTA methods. Thus, even when end-to-end latency is considered, FIT-GNN maintains its efficiency advantage. We will include this total-cost analysis in the revised version.
>
> |Method|Preprocessing|Training|Testing|Model-Agnostic|
> |:---|:---|:---|:---|:---:|
> |**SGGC**|$M+N$  *(CPU)*|$k^2d+kd^2$ *(GPU)*|$N^2d+Nd^2$ *(GPU)*|✓|
> |**GCOND**|$C((N^2+k^2)d+(N+k)d^2)$ *(GPU)*|$k^2d+kd^2$ *(GPU)*|$N^2d+Nd^2$ *(GPU)*|✗|
> |**BONSAI**|$M+N$ *(CPU)*|$k^2d+kd^2$ *(GPU)*|$N^2d+Nd^2$ *(GPU)*|✓|
> |**FIT-GNN**|$M+N$ *(CPU)*|$kd^2+k^2d+\sum_{i=1}^{k}[\bar{n}_i^2d+\bar{n}_id^2]$ *(GPU)*|$\sum_{i=1}^{k}[\bar{n}_i^2d+\bar{n}_id^2]$ *(GPU)*|✓|
>
> > **Table:** Asymptotic time complexities for different condensation methods.
> > **Notation:** $N$ is the number of nodes in the original graph, $M$ is the number of edges, $k$ is the number of condensed nodes, $\bar{n}_i$ is the number of nodes in the $i$-th subgraph where $\bar{n}_i = n_i + \phi_i$, $\phi_i$ is the number of additional nodes appended to subgraph $G_i$, $d$ is the feature dimension, and $C$ is the number of classes.
> > **Note:** (CPU) and (GPU) indicate the hardware used for computation.
>
> We further extend our complexity analysis to the practical scenario, where a new test node $v$ is introduced to the graph $G$. This setting is critical for real-world applications where graphs evolve dynamically. We compare three distinct inference strategies for this new node:
> 1. *Full Graph*: Construct $G_{new}$ by adding $v$ to the original full graph $G$ and infer on $G_{new}$ using the pre-trained network.
> 2. *2nd-hop Neighborhood*: Sample only the 2nd-hop neighborhood of $v$ from $G$ and infer using the pre-trained network (assuming a 2-layer architecture).
> 3. *FIT-GNN Subgraph*: Assign $v$ to a relevant subgraph $G_i$ (e.g., the one containing the majority of its 1st-hop neighborhood) and append the necessary Extra/Cluster nodes. Inference is then performed strictly within the modified subgraph $G_i$.
>
> We present the time complexity comparison for these methods in the table belows.
>
> |Inference Strategy|Preprocessing Complexity|Inference Complexity|Notes|
> |:---|:---|:---|:---|
> |**Full Graph**|$O(1)$|$O(n^2d+nd^2)$|Computationally expensive; requires processing the entire graph for one new node.|
> |**2nd-hop Neighborhood**|$O(\|\mathcal{N}_1(v)\|\Delta^2)$|$O(\|\mathcal{N}_2(v)\|^2d+\|\mathcal{N}_2(v)\|d^2)$|$\|\mathcal{N}_j(v)\|$ is the number of nodes in j-hop neighborhood. $\Delta$ is the maximum degree in graph.|
> |**FIT-GNN Subgraph**|$O(k)$|$O(\bar{n}^2_i d+\bar{n}_id^2)$|Assuming $v$ has maximum neighbors in $G_i$ which has $n_i$ many nodes.|
>
> *Observations*:
> The most significant advantage of the FIT-GNN approach is that inference complexity depends solely on the subgraph size $\bar{n}_i$ ($n_i\ll n$). In contrast, the Full Graph approach remains tied to the global parameter $n$ due to the necessity of loading or accessing the full adjacency matrix.
>
> ---
>
> As a final note, we wish to clarify that the methodology proposed addresses the crucial **inference bottleneck** by enabling GNNs to "fit" in memory during inference, thus opening potential for deploying powerful GNN models on **resource-constrained devices** and on **extremely large graphs**.
>
> We thank the reviewer again for their insightful comments. We are confident that the additional experiments and the clarifications on complexity and notation will significantly strengthen the manuscript, and we are committed to incorporating these changes in the final version.

---

### Review · Reviewer_MkPi · 2026-02-02

**Summary Of Contributions:**

This paper addresses the problem of high inference time and memory consumption in GNNs on large graphs. In contrast to prior work that focuses on reducing training cost, the paper explicitly targets inference efficiency. The proposed FIT-GNN framework applies graph coarsening to partition a graph into subgraphs and performs both training and inference on these subgraphs. To reduce information loss caused by partitioning, the method introduces two strategies, Extra Nodes and Cluster Nodes. The approach is evaluated on node-level and graph-level tasks, including classification and regression.

A key strength of the paper is its clear focus on inference scalability, which is practically important but less studied than training efficiency. The work provides a theoretical analysis of time and space complexity and extensive experiments across many datasets, including very large graphs where baseline methods fail due to memory limits. The results show large reductions in inference time and memory usage while maintaining competitive predictive performance.

A limitation is that the method depends on the choice of coarsening algorithm and coarsening ratio, which introduces additional hyperparameters. In addition, the conceptual difference from earlier partition-based methods could be stated more explicitly to avoid confusion.

**Audience:**

Yes

**Audience Explanation:**

The paper is relevant to researchers and practitioners interested in scalable graph learning and deployment of GNNs in resource-constrained settings.

**Broader Impact Concerns:**

The paper focuses on computational efficiency and does not raise notable ethical concerns.

**Claims And Evidence:**

Yes

**Claims Explanation:**

The claims are supported by both theoretical and empirical evidence.
The paper derives conditions under which subgraph-based inference has lower time and space complexity than standard full-graph inference.
The experimental section evaluates the method on a wide range of benchmarks and tasks, and reports inference time, memory usage, and predictive performance.

**Requested Changes:**

1. The paper would benefit from a clearer and more explicit comparison with Cluster-GCN. While Cluster-GCN is cited, the authors should clearly state that it mainly improves training scalability and still requires full-graph inference, whereas FIT-GNN reduces both training and inference cost. (critical)

2. The complexity analysis is largely presented using dense-matrix formulations and GCN-style computation. A clearer discussion of sparse implementations and how the conclusions extend to other common architectures, such as GraphSAGE or GIN, would improve clarity and practical relevance. (critical)

3. The paper could offer more practical guidance on how to choose the coarsening ratio and coarsening algorithm for new datasets, based on the empirical observations already reported. (non-critical)

4. The presentation of the methodology and analysis could be streamlined to improve readability. (non-critical)
Such as in Section 4.3 and the associated appendices.

---

> ### Author Response · Authors · 2026-02-04
> **Response to Requested Changes**
>
> We thank the reviewers for their insightful feedback and the time invested in evaluating our work. Below, we address the specific changes requested by the reviewers.
>
> # 1. Comparison with Cluster-GCN
> > *Reviewer Comment:* The paper would benefit from a clearer and more explicit comparison with Cluster-GCN... state that it mainly improves training scalability and still requires full-graph inference.
>
> We agree with the reviewer that a direct comparison with Cluster-GCN is essential to position our work correctly. As highlighted by the reviewer, the fundamental difference lies in the inference phase: while Cluster-GCN enables scalable training via subgraph sampling, it typically defaults to full-graph inference, which does not solve the memory bottleneck during deployment. FIT-GNN, by contrast, explicitly targets inference efficiency by processing independent subgraphs.
>
> To empirically demonstrate the difference, we compared accuracy of FIT-GNN against a Cluster-GCN implementation on the Cora, Citeseer, and Pubmed datasets. The results are summarized below:
>
> |Dataset|FIT-GNN|Cluster-GCN (New Baseline)|
> |-|-|-|
> |**Cora**|0.829±0.002|0.785±0.007|
> |**Citeseer**|0.701±0.001|0.694±0.008|
> |**Pubmed**|0.761±0.004|0.774±0.004|
>
> As observed, Cluster-GCN often exhibits a performance drop compared to FIT-GNN (particularly on Cora and Citeseer) due to training on sampled subgraphs rather than the full graph structure.
>
> Furthermore, regarding computational cost, Cluster-GCN does not provide the theoretical time and space complexity reductions during inference that FIT-GNN offers, as it processes the entire graph structure for prediction. We will update the manuscript to explicitly discuss these trade-offs and include these empirical comparisons.
>
> # 2. Complexity Analysis
> > *Reviewer Comment:* The complexity analysis is largely presented using dense-matrix formulations... A clearer discussion of sparse implementations... would improve clarity.
>
> We acknowledge that our initial derivation utilized dense matrix formulations for clarity. In a sparse graph setting, the complexity becomes a function of edges rather than just nodes. However, our core scalability argument holds: for a single-node inference task in FIT-GNN, the inference is constrained to a subgraph. The number of edges in this subgraph ($\bar{m}_i$) is guaranteed to be significantly smaller than the total number of edges in the full graph ($m$). Consequently, the time and space complexity of our approach remains strictly superior to baselines that require the full edge set for propagation, even when using sparse matrix operations.
>
> We expand our discussion to include other propagation functions as requested. We include the time complexity of Single Node Inference task:
> ||Baseline|FIT-GNN|
> |-|-|-|
> |GIN (Dense)|$n^2d+Lnd^2$|$\max(\bar{n}_i)^2d+L\max(\bar{n}_i)d^2$|
> |GIN (Sparse)|$md+Lnd^2$|$\max(\bar{m}_i)^2d+L\max(\bar{n}_i)d^2$ |
> |GraphSAGE (Dense)|$n^2d+nd^2$|$\max(\bar{n}_i)^2d+\max(\bar{n}_i)d^2$|
> |GraphSAGE (Sparse)|$md+nd^2$|$\max(\bar{m}_i)^2d+\max(\bar{m}_i)d^2$|
>
> Here $\bar{n}_i=n_i+\phi_i$ and $\bar{m}_i=m_i+\psi_i$, where $\phi_i$ is the number of additional nodes added in a subgraph and $\psi_i$ is the number of additional edges added in a subgraph. $L$ is the number of layers in MLP (Distinct from GNN Depth) and $d$ is the hidden dimension.
>
> These updates clarify that the efficiency gains of FIT-GNN are robust across different architectures and sparse implementation details.
>
> # 3. Guidance on Coarsening Ratio and Algorithm
> > *Reviewer Comment:* The paper could offer more practical guidance on how to choose the coarsening ratio and coarsening algorithm...
>
> * **Coarsening Ratio:** There is an inherent trade-off; as the coarsening ratio increases, the subgraph size decreases, which improves speed but potentially partitions the graph too aggressively, leading to information loss. Conversely, a smaller coarsening ratio results in larger subgraphs that preserve more information but require more memory. Therefore, we recommend setting the coarsening ratio to the **minimum value for which the largest resulting subgraph fits into the available GPU memory**. This strategy maximizes information retention while satisfying hardware constraints.
>
> * **Coarsening Algorithm:** Based on our ablation studies (Tables 11 and 12 in the appendix), we observed that the `variation_neighborhoods` algorithm delivers the most consistent performance across varying tasks and datasets. We explicitly recommend this as the default choice for new datasets.
>
> # 4. Methodology and Presentation
> > *Reviewer Comment:* The presentation of the methodology and analysis could be streamlined...
>
> We appreciate this feedback. We agree that the description of the methodology was condensed. We will revise Section 4 and the associated appendices to improve the narrative flow and readability, ensuring that the theoretical contributions are more accessible.

---

### Decision · Action_Editor_EWuE · 2026-02-28

**Recommendation:** Accept with minor revision

**Audience:**

Yes

**Audience Explanation:**

The paper addresses a topic relevant to the TMLR community and would likely be of interest to at least a subset of its audience.

**Claims And Evidence:**

Yes

**Claims Explanation:**

This work introduces two graph-coarsening–based inference strategies, Extra Nodes and Cluster Nodes, to improve the scalability of graph neural networks. Unlike prior studies that focus mainly on training efficiency, the method specifically reduces computational cost and memory usage during inference. Experiments on multiple benchmarks show orders-of-magnitude speedups while maintaining competitive performance across classification and regression tasks.

After the rebuttal, two reviewers expressed positive opinions, while one reviewer remained negative mainly due to concerns about novelty. I attached the reviewer’s negative comments: “The submission addresses an important practical problem (reducing GNN inference time and memory), and the idea appears technically plausible. However, in its current form, the claims are not supported by sufficiently clear and convincing evidence to meet the TMLR acceptance bar. Key parts of the paper are difficult to verify due to presentation issues, and its novelty is incremental compared to prior work. Moreover, some findings—especially the large node regression gains—require a more systematic analysis and clearer control of potential confounds. Overall, the combination of incremental novelty, presentation quality, and insufficiently justified experimental evidence limits the interest to the TMLR audience.”

Considering that TMLR places relatively less emphasis on novelty compared to some other venues, I am inclined to recommend acceptance. The authors should provide a more systematic analysis and clearer control of potential confounds regarding the reported “large node regression gains.”